# Isolation and characterization of 24 phages infecting the plant growth-promoting rhizobacterium *Klebsiella* sp. M5al

Marissa R. Gittrich[1,2], Courtney M. Sanderson[1,2], James M. Wainaina[1,2,3], Cara M. Noel[1], Jonathan E. Leopold[1], Erica Babusci[4], Sumeyra C. Selbes[5], Olivia R. Farinas[6], Jack Caine[7], Joshua Davis II[1], Vivek K. Mutalik[8], Paul Hyman[9], Matthew B. Sullivan[1,2,10]*

1 Department of Microbiology, The Ohio State University, Columbus, Ohio, United States of America, 2 Center of Microbiome Science, The Ohio State University, Columbus, Ohio, United States of America, 3 Department of Biology, Woods Hole Oceanographic Institution, Woods Hole, Massachusetts, United States of America, 4 School of the Environment and Natural Resources, The Ohio State University, Columbus, Ohio, United States of America, 5 Department of Psychology, The Ohio State University, Columbus, Ohio, United States of America, 6 College of Public Health, Division of Environmental Health Sciences, The Ohio State University, Columbus, Ohio, United States of America, 7 Department of Biology, Kenyon College, Gambier, Ohio, United States of America, 8 Environmental Genomics and Systems Biology Division, Lawrence Berkeley National Laboratory, Berkeley, California, United States of America, 9 Department of Biology/Toxicology, Ashland University, Ashland, Ohio, United States of America, 10 Department of Civil, Environmental, and Geodetic Engineering, The Ohio State University, Columbus, Ohio, United States of America

* sullivan.948@osu.edu

## Abstract

Bacteriophages largely impact bacterial communities via lysis, gene transfer, and metabolic reprogramming and thus are increasingly thought to alter nutrient and energy cycling across many of Earth's ecosystems. However, there are few model systems to mechanistically and quantitatively study phage-bacteria interactions, especially in soil systems. Here, we isolated, sequenced, and genomically characterized 24 novel phages infecting *Klebsiella* sp. M5al, a plant growth-promoting, nonencapsulated rhizosphere-associated bacterium, and compared many of their features against all 565 sequenced, dsDNA *Klebsiella* phage genomes. Taxonomic analyses revealed that these *Klebsiella* phages belong to three known phage families (*Autographiviridae*, *Drexlerviridae*, and *Straboviridae*) and two newly proposed phage families (Candidatus *Mavericviridae* and Ca. *Rivulusviridae*). At the phage family level, we found that core genes were often phage-centric proteins, such as structural proteins for the phage head and tail and DNA packaging proteins. In contrast, genes involved in transcription, translation, or hypothetical proteins were commonly not shared or flexible genes. Ecologically, we assessed the phages' ubiquity in recent large-scale metagenomic datasets, which revealed they were not widespread, as well as a possible direct role in reprogramming specific metabolisms during infection by screening their genomes for phage-encoded auxiliary metabolic genes (AMGs). Even though AMGs are common in the environmental literature, only one of our phage families, *Straboviridae*, contained AMGs, and the types of AMGs were correlated at the genus level. Host range phenotyping revealed the phages had a wide range of infectivity, infecting between 1–14 of our 22 bacterial strain

**Data Availability Statement:** All 24 phage genomes isolated in this study are available from the NCBI database (accession numbers PP197198,

PP197344, PP202993, PP444685, PP44686
PP44687, PP4688, PP505786, PP505787,
PP526028, PP526029, PP554336, PP554337,
PP554338, PP56864, PP556865, PP582754,
PP582755, PP582756, PP582757, PP582758,
PP597382, PP597383, PP597384.) All phage
stocks are available from the Félix d'Hérelle
Reference Center for bacterial viruses of the
Université Laval or by contacting the
corresponding author Matthew Sullivan at OSU.

**Funding:** This research was supported by the DOE
Office of Science, Office of Biological and
Environmental Research (BER), grants no. DE-
SC0020173 and DE-SC0023307. Marissa R.
Gittrich was supported in part by the NIH Grant
T32 GM086252.

**Competing interests:** The authors have declared
that no competing interests exist.

panel that included pathogenic *Klebsiella* and *Raoultella* strains. This indicates that not all capsule-independent Klebsiella phages have broad host ranges. Together, these isolates, with corresponding genome, AMG, and host range analyses, help build the *Klebsiella* model system for studying phage-host interactions of rhizosphere-associated bacteria.

## Introduction

Phages, viruses that infect bacteria, are key players in the modulation of ecological and evolutionary processes in diverse bacterial communities throughout the Earth's ecosystems [1]. For example, in marine environments, phage infection is responsible for the daily turnover of 20%-40% of marine bacterial biomass and, in turn, impacts nutrient cycling [2, 3]. Additionally, during phage infection, the bacterial host metabolism is drastically reprogrammed through alterations in transcription, protein production, metabolite production, and the import and export of resources that can have diverse impacts on other organisms within the community [4]. Finally, phages drive evolution by transferring genetic material from one bacterium to another via transduction [2]. This can impact essential metabolic functions like photosynthesis, carbon cycling, and nutrient acquisition [5]. Beyond the oceans, phage-bacteria interactions are increasingly studied across many ecosystem types, including the human gut [6, 7], permafrost [8, 9], fracking [10, 11], glacial ice brines [12, 13], and many other environments [1, 14].

The roles of phages in soils are among the most challenging to elucidate due to soil complexity and the lack of cultivated model systems. However, overcoming these challenges is critical to understanding phage impacts in soils. Soil phages are abundant, with phage concentrations ranging from $\sim 10^3$ to $10^9$ particles measured per gram dry weight depending on the soil [15]. Particularly at higher concentrations, it is predicted that there are more frequent phage-bacteria encounters in soils that could lead to higher infection rates than in marine systems [16], and these phage infections are predicted to infect key microbes involved in carbon cycling, which could impact the biogeochemical cycles of these microbes [8, 9]. Hence, phages are now increasingly recognized as major players in soil ecosystems [15, 16].

Problematically, however, there is a lack of cultured and experimentally tested soil phage-host systems that are needed to advance the field from ecological studies to mechanistic hypothesis testing. To date, in terrestrial systems, such work has predominantly focused on phages that target plant bacterial pathogens [17–19], even though phages infecting plant growth-promoting rhizobacteria can drastically impact root microbial communities [18, 20–24]. For example, adding phages to natural and mock soil communities has been found to impact nutrient availability [16, 21], bacterial community structure [22–24], and, in turn, plant health.

*Klebsiella* is an ideal target host for phage model system development as the genus is widespread across soils, freshwater, plants, and humans [25–27]. While 565 *Klebsiella* phages have already been isolated and sequenced [28], these are almost exclusively derived from studies focused on pathogenic *Klebsiella* strains infecting encapsulated strains. No studies have examined phages infecting soil-associated, nonencapsulated *Klebsiella* species. *Klebsiella* sp. M5al has been a model strain for molecular genetics of $N_2$-fixation, production of 1,3-propanediol and 2,3-butanediol, and has been found to colonize rice roots without causing soft rot disease [29–31]. Here, we isolated 24 phages infecting *Klebsiella* sp. M5al. These 24 novel, nonredundant phages were genomically sequenced and compared to 565 complete, genome-sequenced *Klebsiella* phages. Additionally, we examined the host range for these 24 phages by assaying against a panel of *Klebsiella* and *Raoultella* isolates to see if phages isolated on a

nonencapsulated strain can infect multiple species or genera as previously found for other capsule-independent *Klebsiella* phages. Together, these data serve as a framework to expand the understanding of phage-*Klebsiella* interactions within diverse environmental contexts.

## Results and discussion

### Isolation, genomic characterization, and diversity of phages infecting *Klebsiella* sp. M5al

To isolate phages infecting *Klebsiella* sp. M5al, we assembled a collection of 32 samples from soil, water, and sewage around Ohio S1 Table and screened the collection using standard plaque isolation techniques [32] (see Methods). Of the 32 plaquing assays, 14 samples yielded lysis zones, indicating the possible presence of phages. Plaques were isolated and purified from these samples per standard protocol [32], and phage genomes were sequenced. This resulted in 24 novel, nonredundant phages isolated on *Klebsiella* sp. M5al Table 1. The resultant genome sequences revealed a range of genome sizes (38,140–177,853 base pairs), predicted open reading frames (ORFs) (47–310 ORFs), and GC content (38.93%– 56.69%; Tables 1 and S2).

To assign taxonomies, we compared the complete genomes of our 24 isolated phages to all complete viruses in the NCBI virus database [28]. The closest relative was identified via BLAST [31] for each phage and used to calculate intergenomic similarity S3 Table. Taxon rank was assigned according to current guidelines from the International Committee on the Taxonomy

**Table 1. Phage isolate names, taxonomic assignments, and basic genomic characteristics.**

| Phage name | genome size (bp) | #genes | #introns | #tRNAs | Family | Genus | Accession |
|---|---|---|---|---|---|---|---|
| vB_KM5a1-KLB1 | 171972 | 263 | 3 | 1 | *Straboviridae* | *Slopekvirus* | PP444686 |
| vB_KM5a1-KLB3 | 176987 | 282 | 0 | 2 | *Straboviridae* | *Slopekvirus* | PP444687 |
| vB_KM5a1-KLB10 | 174281 | 282 | 3 | 2 | *Straboviridae* | *Slopekvirus* | PP505786 |
| vB_KM5a1-KLB13 | 174344 | 277 | 4 | 2 | *Straboviridae* | *Slopekvirus* | PP505787 |
| vB_KM5a1-KLB15 | 174800 | 277 | 1 | 2 | *Straboviridae* | *Slopekvirus* | PP526028 |
| vB_KM5a1-KLB21 | 177853 | 278 | 5 | 2 | *Straboviridae* | *Slopekvirus* | PP597383 |
| vB_KM5a1-KLB23 | 175398 | 278 | 3 | 2 | *Straboviridae* | *Slopekvirus* | PP554337 |
| vB_KM5a1-KLB25 | 174636 | 278 | 3 | 2 | *Straboviridae* | *Slopekvirus* | PP556864 |
| vB_KM5a1-KLB27 | 176864 | 276 | 4 | 2 | *Straboviridae* | *Slopekvirus* | PP582754 |
| vB_KM5a1-KLB2 | 165103 | 286 | 3 | 17 | *Straboviridae* | *Jiaodavirus* | PP597384 |
| vB_KM5a1-KLB8 | 165787 | 290 | 3 | 17 | *Straboviridae* | *Jiaodavirus* | PP444688 |
| vB_KM5a1-KLB31 | 176609 | 310 | 5 | 6 | *Straboviridae* | *Kanagawavirus* | PP582757 |
| vB_KM5a1-KLB19 | 52663 | 88 | 1 | 1 | *Drexlerviridae* | *Vilniusvirus* | PP526029 |
| vB_KM5a1-KLB26 | 51740 | 85 | 1 | 0 | *Drexlerviridae* | unclassified | PP556865 |
| vB_KM5a1-KLB4 | 38788 | 49 | 0 | 0 | *Autographiviridae* | *Teetrevirus* | PP197344 |
| vB_KM5a1-KLB7 | 38140 | 48 | 0 | 0 | *Autographiviridae* | *Teetrevirus* | PP582758 |
| vB_KM5a1-KLB12 | 38195 | 47 | 0 | 0 | *Autographiviridae* | *Teetrevirus* | PP597382 |
| vB_KM5a1-KLB16 | 44851 | 70 | 0 | 0 | Ca. *Mavericviridae* | *Alumvirus* | PP197198 |
| vB_KM5a1-KLB22 | 45183 | 72 | 0 | 0 | Ca. *Mavericviridae* | *Buckeyevirus* | PP554336 |
| vB_KM5a1-JVSB2 | 47362 | 65 | 0 | 0 | Ca. *Mavericviridae* | *Ashvirinae* | PP444685 |
| vB_KM5a1-KLB5 | 39747 | 60 | 0 | 0 | Ca. *Rivulusviridae* | *Darbyvirus* | PP202993 |
| vB_KM5a1-KLB24 | 40194 | 53 | 0 | 0 | Ca. *Rivulusviridae* | *Sherbvirus* | PP554338 |
| vB_KM5a1-KLB28 | 40269 | 55 | 0 | 0 | Ca. *Rivulusviridae* | *Sherbvirus* | PP582755 |
| vB_KM5a1-KLB29 | 40017 | 53 | 0 | 0 | Ca. *Rivulusviridae* | *Colbvirus* | PP582756 |

Phage naming conventions and taxonomic assignments were made using current best practices (see Methods).

of Viruses (ICTV; see methods; 32). Using these standards, 16 of the 24 phages were assigned to four existing genera spanning three families: *Autographiviridae*, *Drexlerviridae*, and *Straboviridae* Table 1. One phage, KLB26, was unclassified at the family level but had 59% intergenomic similarity to a *Drexlerviridae* phage S3 Table. To confirm that KLB26 belongs to a novel genus in the *Drexlerviridae* phage family, we used core genome analyses, gene-sharing network family prediction [33], and phylogenetic comparisons S4 and S5 Tables [32]. Finally, seven phages could not be classified into existing taxa. Using gene-sharing networks [33] and core genome clustering, these phages were found to belong to two novel phage families that we are proposing: Candidatus (Ca.) *Mavericviridae* (n = 3) and Ca. *Rivulusviridae* (n = 4).

## Occurrence of the isolated phages in soils and rivers

Since these phages were isolated from soils, rivers, and sewage, we aimed to determine whether they occurred in recently available large-scale metagenomic samples from soil or freshwater environments. To this end, we screened our phage genomes against the Global Soil Virus Atlas (GSVA) [34], which contains virus genome fragments identified from 2,953 sequenced soil metagenomes and the Genome Resolved Open Watersheds database (GROWdb) [35], where we identified virus genome fragments from the 163 surface water riverine samples that offer representation of approximately 90% of the U.S. watersheds [35].

Due to privacy restrictions on the part of the underlying GSVA data and the recent availability of the GROWdb dataset, we were unable or hesitant, respectively, to use the original dataset reads for read-mapping-based detection. Instead, we asked whether the genomes of our 24 phages or the genome sequencing reads from our 24 phages shared significant similarities to the uncultivated virus genomes (UViGs) from the GSVA and GROWdb datasets. This revealed that none of our genomes showed significant similarity to the uViGs in either dataset but that at least some minimal number of reads from three of the phage genomes–KLB19 (0.004%), KLB22 (0.02%), and KLB29 (0.05%)–mapped to three sample sites in the GSVA dataset (S6 Table), though notably, these uViGs were less than 10kb and low quality, leading to their exclusion after quality control [34]. Thus, our phage isolates do not appear to represent widespread ecological genotypes, at least in these soil and river datasets.

However, we posit that this limited detection is due to the high diversity of soil viruses, which are often unique to specific environments even when sampled just mere meters apart [34, 36–38]. Indeed, within the GSVA dataset, only 13.9% of (vOTUs) were found in more than one sample, and fewer than 1% were present in more than five samples [34]. This suggests that the soil and river virology community has many new taxa, phenomena, and biological mysteries remaining to be explored.

## Genomic diversity of newly isolated phages compared to all known *Klebsiella* phages

Next, we explored how our newly isolated and sequenced phage genomes compare to previously sequenced *Klebsiella* dsDNA phage genomes. These phages were predominantly derived from *Klebsiella* strains that are both capsule producers and human pathogens [39, 40]. As of February 6th, 2023, there were 565 nonredundant dsDNA phages with complete genomes in the NCBI Virus [28]–within the class *Caudoviricetes* S7 Table. To compare our phages to all known *Klebsiella* phages, we created a concatenated phylogenetic tree using the large terminase, small terminase, portal protein, and major capsid protein from representatives of the 565 nonredundant phages (defined as having <80% coverage and <95% percent identity to another phage) (Fig 1, see Methods). These analyses revealed that of the 523 representative *Klebsiella* phage genomes, 441 assort into nine ICTV-recognized phage families [41], and 124

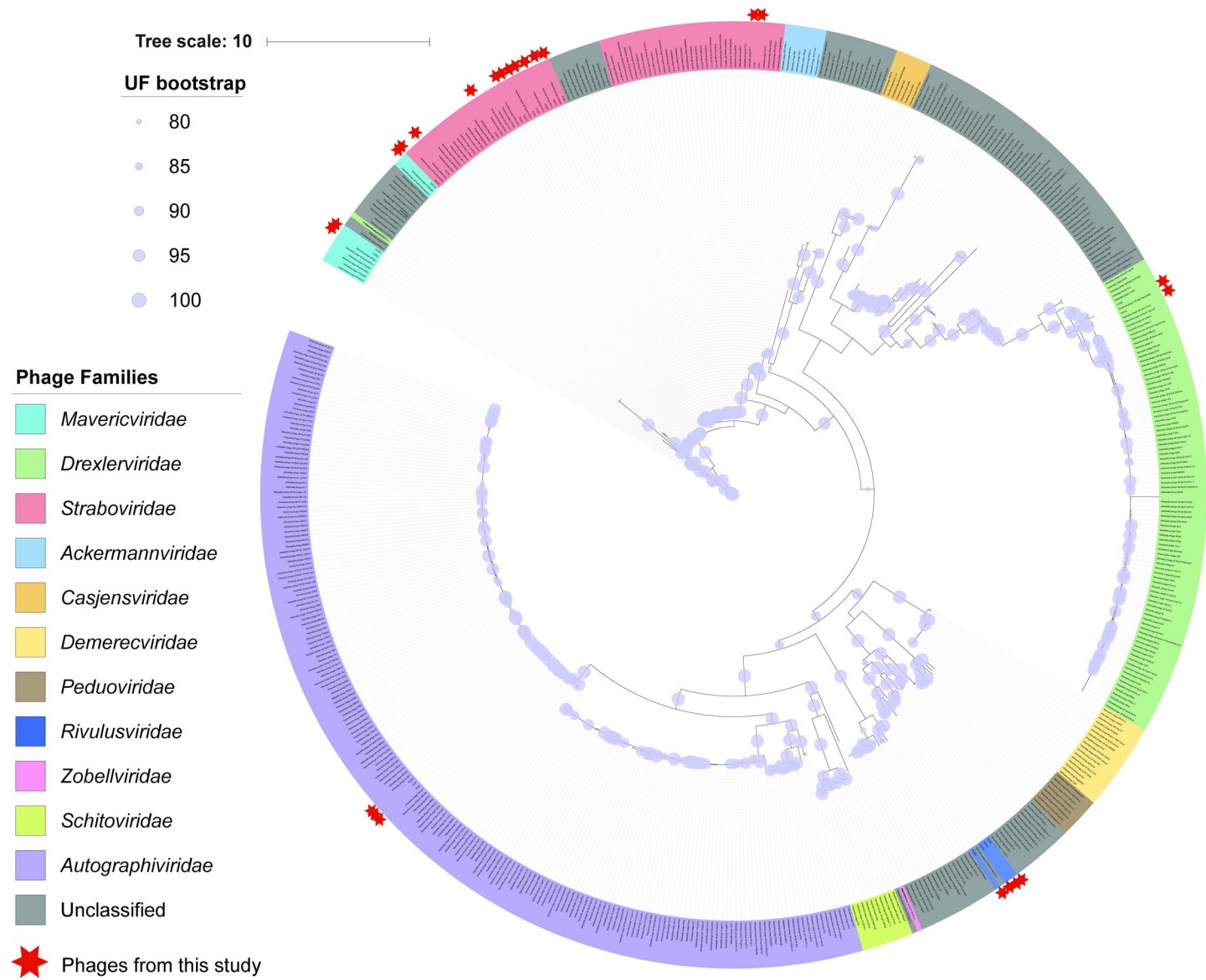

**Fig 1. Phylogenetic tree of *Klebsiella* phages.** Phylogenetic tree of 523, *Klebsiella* infecting, NCBI representative phage genomes, and the phages isolated in this study. The partitioned tree was constructed using concatenated sequences from the small terminase, large terminase, portal, and major capsid proteins. Phage names are colored according to their assigned phage family, and phages isolated in this study are marked by red stars. UF bootstrap represents the ultra-fast bootstrap values calculated by IQTree. The tree scale refers to the number of amino acid substitutions per site.

were unclassified at the family level. Most (64.6%) belonged to three phage families [*Autographiviridae* (n = 210), *Drexlerviridae* (n = 98), and *Straboviridae* (n = 57)], with minimal representation (11.1%) from the remaining six families [*Dermercviridae* (n = 17), *Peduoviridae* (n = 16), *Schitoviridae* (n = 10), *Ackermannviridae* (n = 10), *Casjensviridae* (n = 9), and *Zobellviridae* (n = 1)].

With this taxonomic framework from published phages, we next sought to assess where our phages taxonomically belonged. This revealed that the 24 phages belonged to five families, including 17 from the three largest phage families [*Autographivridae* (n = 3), *Drexlerviridae* (n = 2), and *Straboviridae* (n = 12)], and seven that assorted into two novel families as described above, Ca. *Mavericviridae* (n = 3) and Ca. *Rivulusviridae* (n = 4) (Fig 1). Given these

taxonomic assignments and a diverse phage collection that infects nonencapsulated *Klebsiella*, we next sought to characterize the host range and genomic content of the phages to assess signatures of niche differentiation within and between these families.

**Straboviridae family.** The *Straboviridae* phage family originally belonged to the *Myoviridae* phage family until ICTV updates in 2022, which separated the T4-like phages into the *Straboviridae* infecting *Gammaproteobacteria* and the *Kyanoviridae* infecting cyanobacteria [41]. Focusing only on the former, there are currently 633 *Straboviridae* representatives, 57 of which infect *Klebsiella* species. Their genome sizes range from 59 to 349 kb, with an average genome size of 172.4 kb S8 Table [28]. These 57 *Klebsiella* infecting *Straboviridae* phages represent three phage genera, *Jiaodavirus* (n = 32), *Slopekvirus* (n = 22), and *Pseudotevenvirus* (n = 1), and two phages unclassified at the genus level S8 Table.

Our study isolated 12 *Straboviridae* phages from sewage, water, and soil that belong to three phage genera, *Jiaodavirus* (n = 2), *Kanagawavirus* (n = 1), and *Slopekvirus* (n = 9) Tables 1 and S1. The genome sizes of our 12 *Straboviridae* phages range from 165.7–177.8 kb with 262–310 predicted open reading frames (ORFs) and 1–17 tRNAs Table 1 (Figs 2A, 2B and S1). The phage-encoded tRNAs were nearly identical within all phages isolated in each genus (Fig 2A). We compared codon usage between *Klebsiella* sp. M5al to see if tRNA presence was due to codon usage or amino acid usage bias, comparing all ORFs and predicted late ORFs [42]. No correlation existed between phage codon or amino acid usage and tRNAs encoded by the phage S1 Fig. However, this may not be surprising, as studies of codon usage bias in other phages have had mixed results [42–50].

**Drexlerviridae family.** The *Drexlerviridae* family originally belonged to the *Siphoviridae* phage family until ICTV updates in 2020 [28]. This phage family has 367 complete genome-sequenced phages infecting *Gammaproteobacteria* (n = 345) and *Campylobacterota* (n = 22), including 98 that infect *Klebsiella*. The genome sizes range from 22–78 kb. Of the 98 *Klebsiella Drexlerviridae* phages, 92 derive from the phage genus *Webervirus*, one from the *Vilniusvirus* genus, and five are currently unclassified at the genus level S9 Table.

Our study isolated two *Drexlerviridae* phages, KLB26 and KLB19, from 2 environmental river water samples S1 Table. KLB19 belongs to the genus *Vilniusvirus and* has a phage genome length of 52.6 kb consisting of 88 ORFs, including one tRNA for serine and an intron within the tape measure chaperone Table 1 (Fig 2C). Like the *Straboviridae* phage family, there was no correlation between phage codon and amino acid usage for KLB19 S3 Fig. KLB26 is unclassified at the genus level, has a genome length of 51.7 kb, and comprises 86 ORFs with an intron in the tape measure chaperone Table 1 (Fig 2C).

**Autographiviridae family.** The *Autographiviridae* phage family originally belonged to the *Podoviridae* family until 2020 ICTV updates [28]. The *Autographiviridae* phage family comprises 1,106 complete genomes with genome sizes from 30 kb - 61 kb. The *Autographiviridae* phages infect six classes of bacteria: *Alphaproteobacteria* (n = 63), *Bacilli* (n = 2), *Betaproteobacteria* (n = 49), *Cyanophyceae* (n = 44), *Gammaproteobacteria* (n = 947), and *Myxococcota* (n = 1). Of the phages that infect *Gammaproteobacteria*, 210 infect *Klebsiella* and represent five phage genera *Przondovirus* (n = 100), *Drulisvirus* (n = 78), *Teetrevirus* (n = 14), *Ningirsuvirus* (n = 2), *Eapunavirus* (n = 1), and 14 phages unclassified at the genus level S10 Table.

Our study isolated and sequenced three *Klebsiella* phages—KLB4, KLB7, and KLB12—in the genus *Teetrevirus* from two sewage plants S1 Table. The genome sizes of our 12 *Straboviridae* phages range from 38.1–38.7 kb with 47–49 predicted ORFs. KLB7 and KLB12 belong to the same phage species and have 99.9% intergenic similarity. The differences include one single nucleotide polymorphism across the 38kb genomes, and KLB12's genome is 55bp larger than KLB7's genome (in a non-coding region of the phage genome; Fig 2D). KLB4 belongs to

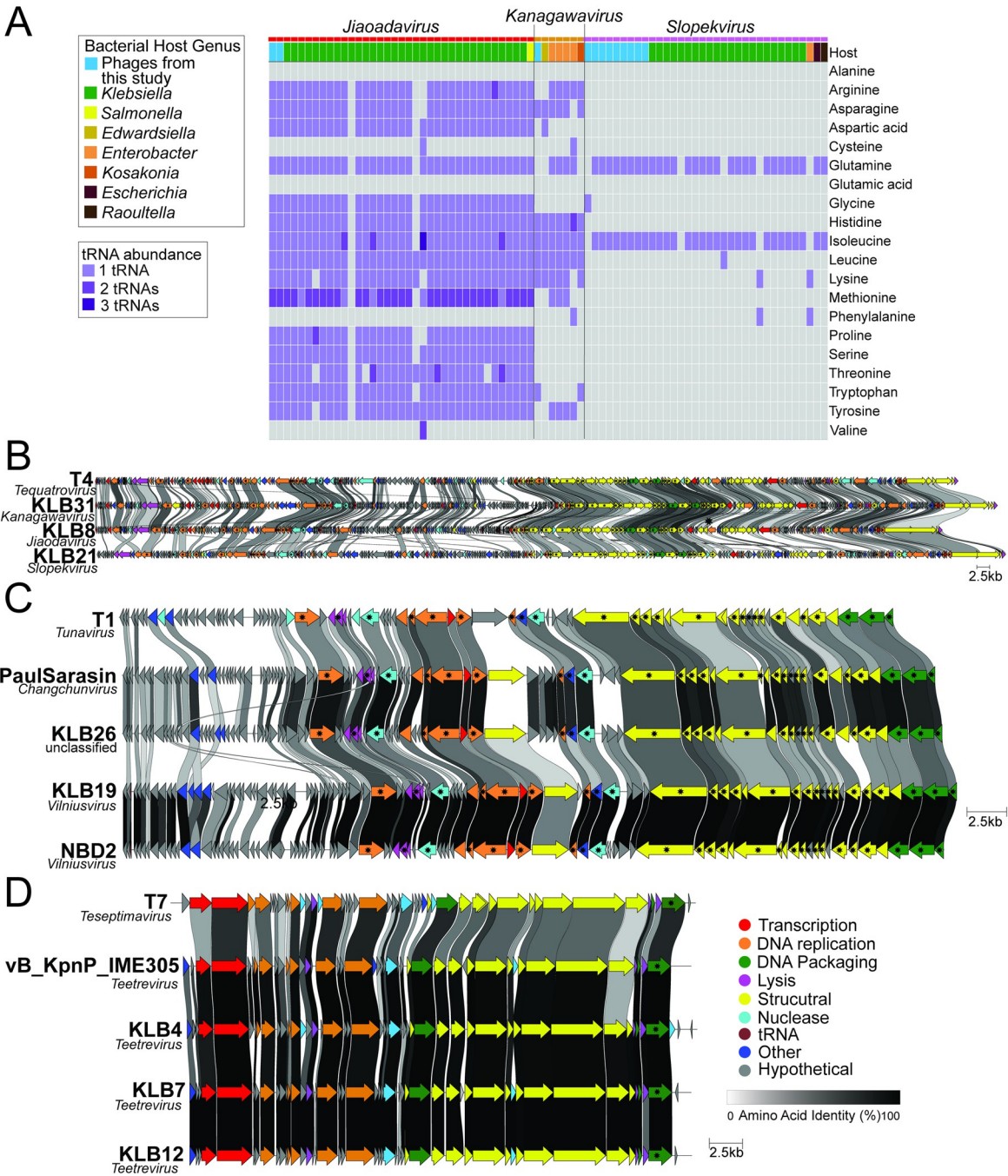

**Fig 2. Genome comparisons of *Klebsiella* phages isolated in this study.** (A) tRNA abundances of all *Straboviridae* phages in the *Kanagawavirus*, *Jiaodavirus*, and *Slopekvirus* genera. Genome comparisons of *(B) Straboviridae* (C) *Drexlerviridae* (D) *Autographiviridae* phages to model phages in each family. Arrows represent forward (right) or reverse (left) open reading frames (ORFs), ORFs are color-coded by function, and ORFs shared across genomes are connected by shading that denotes their percent identity. Core ORFs shared across all RefSeq genomes in this family are marked with a black star. The genome figure was made using Clinker. S2 Fig. shows the *Straboviridae* genome broken into four parts for a zoomed-in comparison of the phage genomes.

a different phage species (∼89% intergenomic similarity to KLB7 and KLB12), with 11 unique or significantly divergent ORFs from KLB7 and KLB12 (Fig 2D).

**Ca. *Mavericviridae*.**  Three phages—JVSB2, KLB16, and KLB22—were unclassified at the family level based on BLAST S3 Table [31]. These phages were isolated from soil, water, and sewage S1 Table and had genome sizes ranging from 44.8 to 47.3 kb with 65–72 ORFs Table 1 (Fig 3A).

Comparing these phages to all NCBI viruses with complete genomes using BLAST [51] and gene-sharing networks [33], we identified 337 phages clustered into a single family. However, based on core genome and phylogenetic analyses, these 337 phages were reduced to 95, which shared core genes found in >95% of all phages and formed a monophyletic lineage.

We sought to determine if these phages were a novel phage family. Core genome analysis of the 95 phages identified ten genes present in at least 95% of the phages S11 Table. We used ViPTree [52] to generate a phylogenetic tree based on the viral proteome, which revealed that these phages represent a monophyletic group and are closely related to *Drexlerviridae* S4 Fig. To determine if this group of phages could be a novel family or belonged to, *Drexlerviridae*, we ran core genome analyses using all 95 phages and all *Drexlerviridae* phages. We identified no core genes shared between *Drexlerviridae* and the Ca. *Mavericviridae* S12 Table and found that Ca. *Mavericviridae* showed a monophyletic lineage, supporting this group as a novel family (See methods; Fig 3B). With these 95 phages, we further evaluated taxonomic ranks based on ICTV recommendations. Genomes with intergenomic similarity ≥70% were classified in the same genus and ≥95% in the same species [53]. Based on the intergenomic similarity calculated via VIRDIC [54], we propose 11 genera; our phages belong to three genera: Ca. *Ashvirus*, Ca. *Buckeyevirus*, and Ca. *Alumvirus* S5 Fig.

**Ca. *Rivulusviridae*.**  Four phages–KLB5, KLB24, KLB28, and KLB29 –were unclassified at the family level based on BLAST. These phages were isolated from soil and water and had genome sizes ranging from 39.7–40.2 kb with 53–60 ORFs Table 1 (Fig 4A). KLB24 and KLB28 share 96.4% intergenomic similarity. KLB29 is more closely related to KLB24 than KLB28, with 64.1% and 65.2% intergenomic similarities, respectively. KLB5 is distantly related to KLB24, KLB28, and KLB29 with 5.1–5.6% intergenomic similarity S6 Fig.

These phages had no close relatives found via BLAST [51], but gene-sharing networks [33] identified 17 phages predicted to be related at the family level. Core genome analyses found 21 core genes. These core genes formed a monophyletic lineage using all proteomes in VipTree and through phylogenetic analyses of the core genes of Ca. *Rivulusviridae* (Figs 4B and S4, see methods), further supporting a novel phage family. With these 17 phages, we propose five phage genera based on intergenomic similarity [54], with our phages belonging to three: *Colbvirus*, *Sherbvirus*, and *Darbyvirus* S6 Fig.

## Core genes of phages

With recent ICTV shifts to assigning phage families based on genetic similarity and core genes instead of phage morphology [41], we examined family-level core gene sets for each phage family. For previously identified phage families, we compared our phage genomes to RefSeq phages [*Straboviridae* (n = 222), *Drexlerviridae* (n = 122), *Autographiviridae* (n = 376)] and for the novel phage families with significantly fewer RefSeq phages we used all RefSeq and non-RefSeq phages [*Mavericviridae* (n = 96) and *Rivulusviridae* (n = 17)]. To account for the impact of misannotation or poor-quality sequences, we looked at the soft-core genome, defined as the core genes found in >95% of all phage genomes [55–57]. These analyses revealed that, on average, 18% of phage genes were core genes. The number of core genes in each phage family ranges from 1 to 43 genes, corresponding to 2 to 36.4% of genes. The majority of these genes were structural proteins that are part of the capsid and tail structures (S4 and S11–15 Tables and Fig 5).

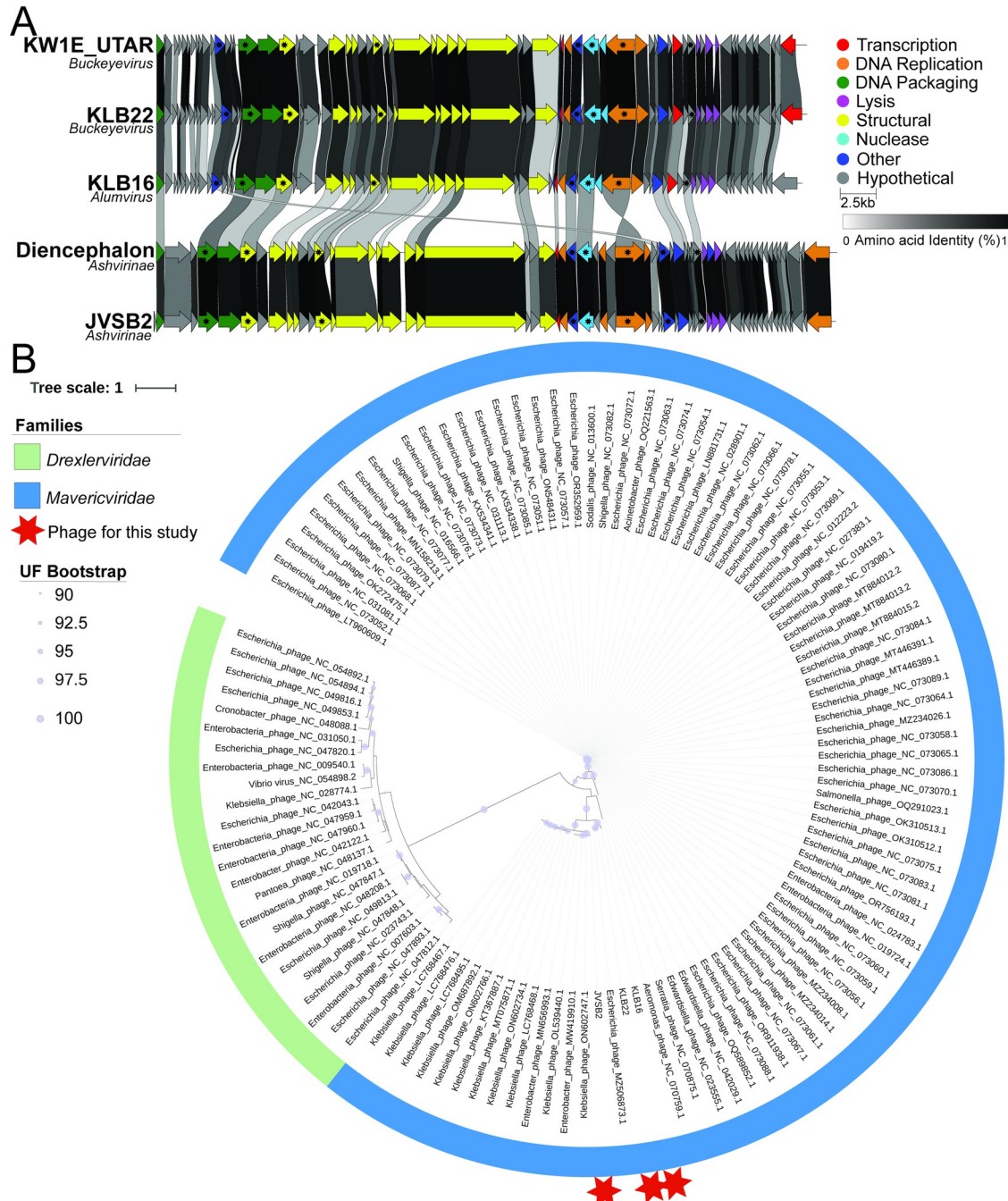

**Fig 3. Genome comparisons of Ca. *Mavericviridae* phages.** (A) Phage genome comparison of our Ca. *Mavericviridae* phages against the closest reference genomes. Arrows represent forward (to the right) or reverse (to the left) ORFs, ORFs are color-coded by function, and ORFs shared across genomes are connected by shading that denotes their percent identity. Core ORFs shared across all RefSeq genomes in this family are marked with a black star. The genome figure was made using Clinker. (B) Concatenated phylogenetic tree of the ten core genes found in >95% of all Ca. *Mavericviridae* phage families. Red stars denote the phages isolated in this study. UFBootstrap support is shown as a number on each node. The tree scale refers to the number of amino acid substitutions per site.

The *Straboviridae* family had the most functionally diverse core genes, with 15.7% of the ORFs being core genes (n = 43). These genes include transcription, DNA replication, structural, DNA packaging, nucleases, and other functions such as polynucleotide kinase, metallo-

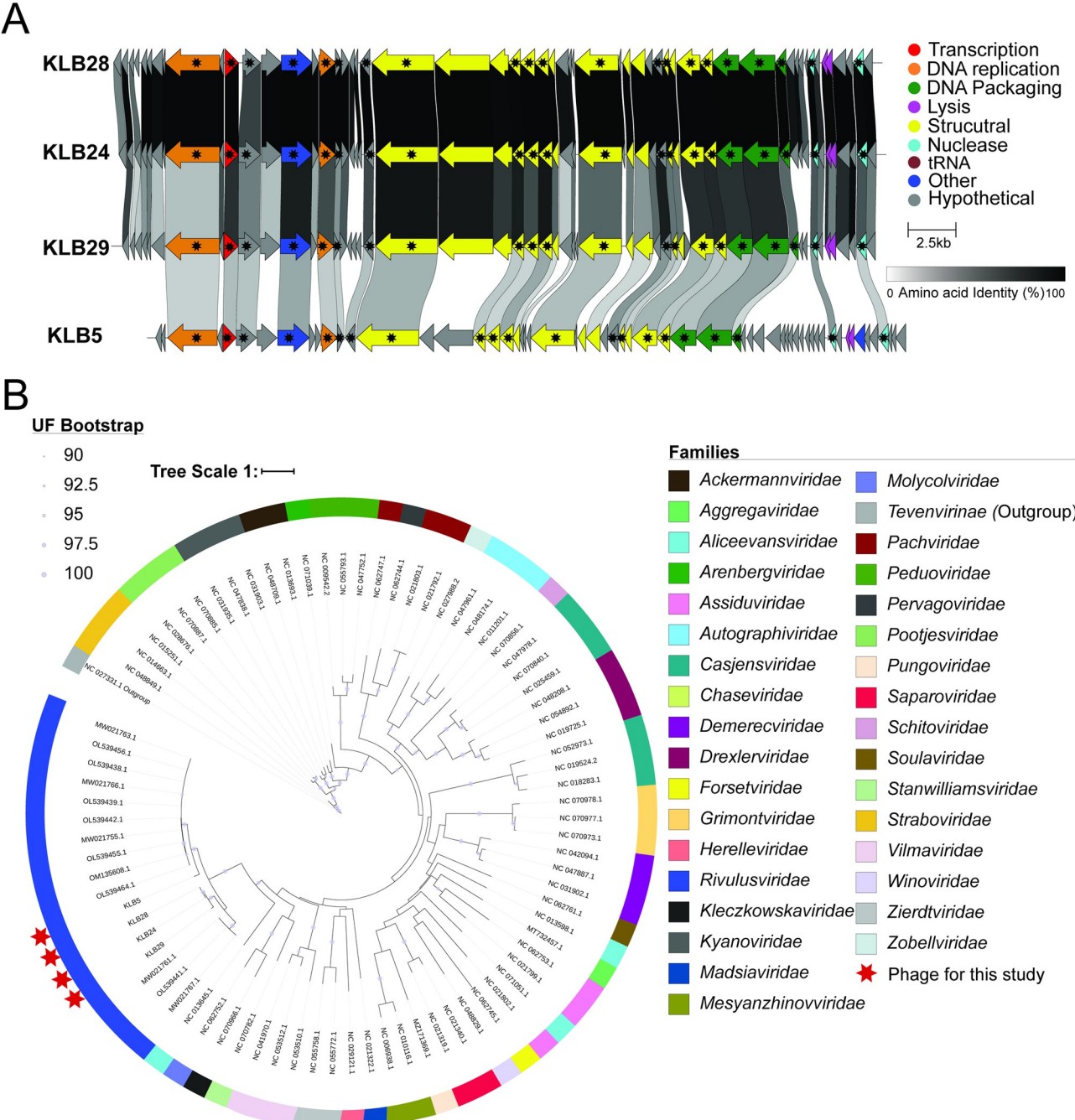

**Fig 4. Phage genome comparison of Ca. *Rivulusviridae* phages.** (A) Arrows represent forward (to right) or reverse (to left) ORFs, ORFs are color-coded by function, and ORFs shared across genomes are connected by shading that denotes their percent identity. Core ORFs, genes shared across all RefSeq genomes, in this family are marked with a black star. The genome figure was made using Clinker. (B) Phylogenetic tree of the large terminase of representatives of each dsDNA family compared to the Ca. *Rivulusviridae* phages. Red stars denote the phages isolated in this study. UFBootstrap support is shown as a circle on each node. The tree scale refers to the number of amino acid substitutions per site.

phosphoesterase, and an exonuclease (Fig 5A), and many of these genes have previously been identified when examining T4-like phages [58–62]. The *Straboviridae* phage family encoded two core genes involved in transcription, a late transcription coactivator and a late sigma

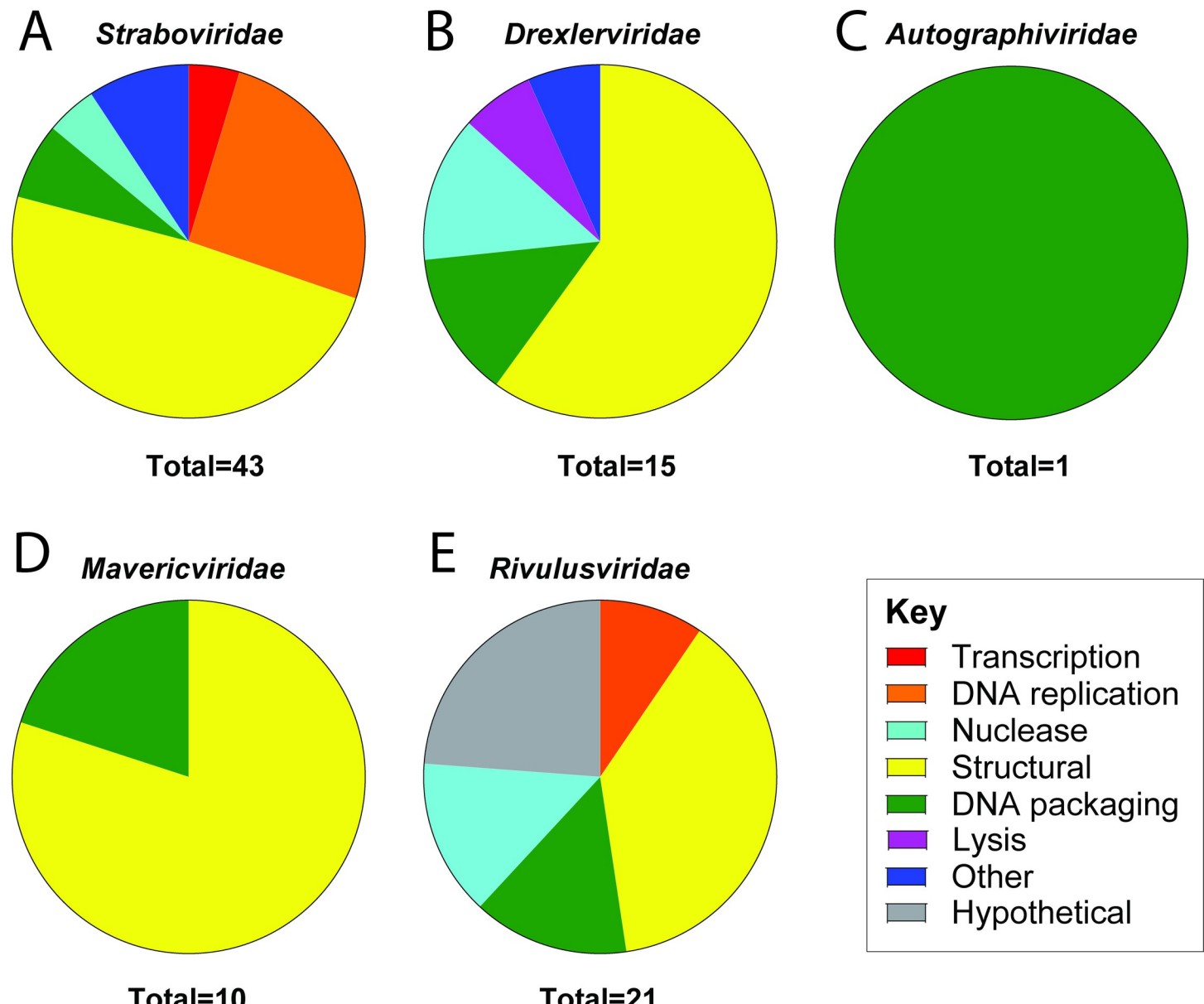

**Fig 5. Core genes for each phage family.** Core genes of the (A) *Straboviridae*, (B) *Drexlerviridae*, (C) *Autographivirdae*, (D) Ca. *Mavericviridae* and (E) Ca. *Rivulusviridae* phage families.

transcription factor, to guide the host RNA polymerase to genes involved in the late stages of phage infection. In contrast, none of the other phage families transcription factors were core genes. This is likely due to the phage family encoding its machinery for DNA replication and the larger phage genome compared to the other phage families. However, in *Escherichia coli* T4, ~1/3 of the ORFs, >100 genes, are involved in the takeover of the host RNA polymerase and transcription during early, middle, and late gene expression [63, 64]. Even with the number of genes involved in host takeover presumably, the lack of shared transcription- and replication-associated core genes indicates host-interaction strategies that more strongly adapt to individual hosts.

The remainder of the phage families presented much simpler cases. First, *Drexlerviridae* had 15 core genes, representing 19.2% of all ORFs (Fig 5B). Unlike *Straboviridae*, no genes are involved in DNA replication; this is unsurprising since *Drexlerviridae* phages, like T1, rely on bacterial host machinery for DNA replication and transcription [65]. Second, *Autographiviridae* had one core gene (Fig 5C), the large subunit terminase involved in DNA packaging. While *Autographiviridae* was initially proposed partly due to the conserved large, single-subunit RNA polymerase responsible for middle and late phage transcription, our analyses did not find that as a core gene. MMseqs2 [66] generated 17 unique clusters for the single-subunit RNA polymerase. Additionally, protein-sharing networks using all proteins show low protein sharing between some *Autographiviridae* subfamilies [67]. These findings indicate that further evaluation for this phage family is likely needed. Third, Ca. *Mavericviridae* had ten core genes, representing 16.9% of all ORFs (Fig 5D). These genes were involved in DNA packaging and structural proteins. Finally, Ca. *Rivulusviridae* had 21 core genes, representing 36.4% of all ORFs (Fig 5E). These genes were involved in DNA replication, structural, DNA packaging, nucleases, and hypothetical proteins. One of those core genes was a bifunctional DNA primase and polymerase that is predicted to synthesize DNA directly from dNTPs without a DNA primer bound to the DNA template. This class of primase and polymerase has previously been found in archaea, bacteria, humans, and some bacteriophages [68].

## Auxiliary metabolic genes across the *Klebsiella* phages

When phages infect a bacterial host, they can drastically reprogram bacterial metabolism toward producing phage proteins [4]. Though challenging to *in silico* predict such reprogramming from genomes alone, some phages encode auxiliary metabolic genes (AMGs) that represent bacterial-derived genes that help redirect energy and resources toward phage production [69] and have been the focus of >100 papers in the viral ecology literature because of their prominence in environmental phages [70]. Thus, AMGs can indicate metabolic 'knobs' that phages tune to maximize phage production under varied conditions of phage-host co-evolution and ecological niche differentiation.

To identify AMGs, we screened our phages using current standard, scalable procedures emerging for virus ecogenomics [70–72]. This identified 48 AMGs found in 12 of our 24 phages. These AMGs were only found in the *Straboviridae* family. Among these 48 AMGs, predicted functional roles range from amino acid biosynthesis to NAD recycling and nucleotide metabolism (Fig 6A and 6B).

Since all these AMGs were found in a single phage family, we screened the *Klebsiella* phage dataset to see if AMG presence was family-dependent. Among the nine ICTV classified families, four of the families had the majority of the phages encoding AMGs. One family, *Casjensviridae*, exhibited AMGs in five out of nine phages, and four showed no AMGs (Fig 6C). We further explored potential family trends among unclassified phages by clustering them at the family level using a gene-sharing network. Among the 18 family clusters identified, four families exclusively encoded AMGs, seven had none, and seven displayed a mix of phages with and without AMGs S16 and S17 Tables (Fig 6C). To gain insights into why certain phage families contained AMGs, we assessed the presence and abundance of AMGs relative to genome size. Interestingly, phages with AMGs tended to have significantly larger genomes (p = <0.0001). Additionally, the number of AMGs was correlated with genome size ($R^2$ = 0.28 p = <0.0001) S7 Fig. This analysis elucidates a nuanced distribution pattern of AMGs across *Klebsiella* phage families. It suggests a potential association between genome size and AMG presence, highlighting the intricate nature of viral genomic content.

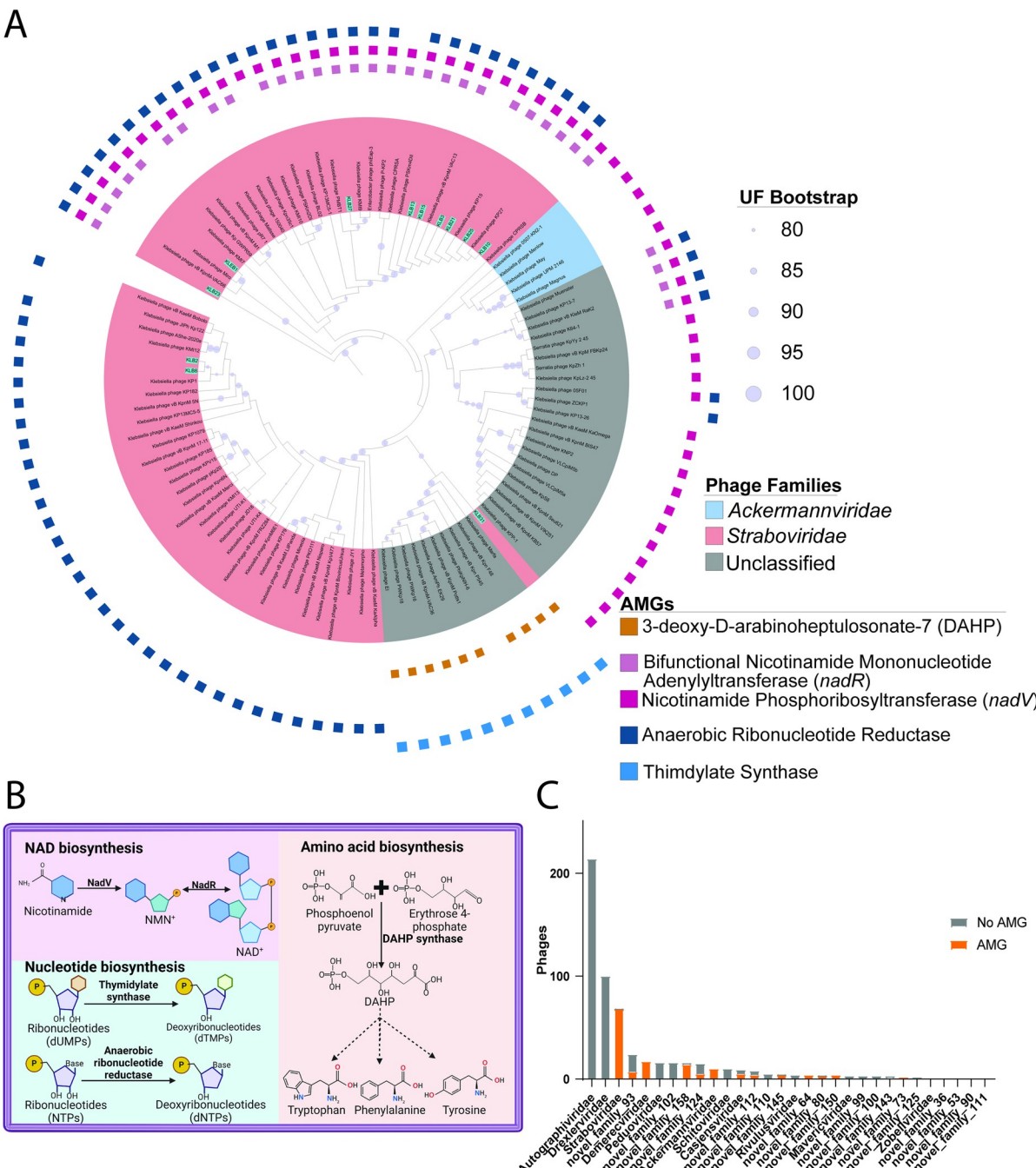

**Fig 6. AMGS found in the *Straboviridae* phages.** (A) Phylogenetic tree of five of the types of AMGS identified in *Klebsiella* phages. The inner ring is colored by the phage family. Outer rings are colored by the AMG type identified using Dram-v. This tree was pruned from the tree in Fig 1. (B) Graphical representation of the AMGs identified in the phages isolated in this study created with BioRender.com. (C) Bar graph showing the number of phages in each family that had an AMG or did not have an AMG. The unclassified phages were clustered into family groupings using a gene-sharing network.

Beyond family-level AMG carriage observations, we next explored what kinds of functions *Klebsiella* phage genomes encode via AMGs. Among the *Straboviridae*, our phages encode several genes involved in nucleotide biosynthesis. Viruses are well known for reprogramming

nucleotide metabolism, as nucleotides are necessary for DNA replication and transcription. It has been debated whether nucleotide biosynthesis genes should be classified as AMGs due to their involvement in common viral functions [73]. However, these nucleotide biosynthesis AMGs help us understand how phages are reprogramming the metabolic potential of the bacterial host and serve an essential role in increasing the host's metabolic capacity to aid in nucleotide production for phages with high nucleotide demands. To illustrate this, our phages in the genera *Slopekvirus* and *Jiaodavirus* encoded an anaerobic ribonucleotide reductase that reduces ribonucleotides to 2'-deoxyribonucleotides which can then be used as a precursor for dNTPs [74]. The *Kanagawavirus* KLB31 encodes a thymidylate synthase that catalyzes the synthesis of dTMP [75]. We hypothesize that these phage-specific AMG differences indicate two unique approaches to acquiring nucleotides during DNA replication, either through ribonucleotides or via the thymidine metabolic route.

Beyond nucleotide-related AMGs, we also found amino acid-related AMGs. KLB31 encodes a 3-deoxy-D-arabinoheptulosonate-7-phosphate (DAHP) synthetase. This enzyme catalyzes the first step in the seven-step shikimate pathway for the biosynthesis of three aromatic amino acids (phenylalanine, tyrosine, and tryptophan), effectively controlling the amount of carbon entering the pathway [76]. Though not yet observed in terrestrial phages, genes related to the shikimate pathway were recently identified in a global oceans AMG study [70]. We hypothesize that during infection, these AMGs may help generate aromatic amino acids or products that use these amino acids, such as precursors for other metabolites, such as those involved in electron transport, communication, cofactors, and lipid production [77].

Finally, we identified AMGs associated with phages potentially affecting redox balance. Specifically, all *Klebsiella Slopekvirus* genus phages encode enzymes involved in salvaging nicotinamide adenine dinucleotide (NAD), which plays crucial roles in electron transfer and various metabolic processes during bacterial and phage infection [78, 79]. These enzymes facilitate the conversion of nicotinamide to NAD+, essential for nucleotide biosynthesis and modifying transcription and DNA synthesis during lytic phage infection [80, 81]. This suggests that phages with NAD+ salvaging pathways utilize NAD+ for multiple functions, including metabolic processes and modifying host RNA polymerase activity to favor phage promoters.

## Host range of phages

Many *Klebsiella* isolates are covered by a capsule, a thick and diverse polysaccharide matrix surrounding the bacterium that masks cell surfaces [82]. *Klebsiella* phages have evolved a depolymerase to digest the capsule and reach the cell surface [83]. These capsules are highly diverse, with >79 capsular serotypes [82]. As such, phages recognize specific serotypes, which results in many *Klebsiella* phages infecting only a narrow range of hosts [84, 85]. Phages that do not rely on the capsule for adsorption have been found to have a broader host range, even against encapsulated *Klebsiella* isolates [84, 85]. Since our phages infect a nonencapsulated *Klebsiella* and do not rely on the capsule for adsorption, we sought to see if our phages had broader host ranges.

We performed host-range analyses for the 24 phages against 22 bacterial isolates that represented 4 *Klebsiella* species (19 isolates) and 3 *Raoultella* species closely related to *Klebsiella* (3 isolates) S18 Table (Fig 7). The phages with the broadest host range were the *Straboviridae* and *Autographiviridae* phage families, while the Ca. *Mavericviridae*, KLB5 and KLB26, had the narrowest host ranges, infecting 1–2 *Klebsiella* or *Raoultella* isolates (Fig 7). This indicated that not all phages isolated on nonencapsulated *Klebsiella* have broad host ranges. The host range of specific phages may be due to other phage and host genomic characteristics such as tail receptors, phage defense systems, etc.

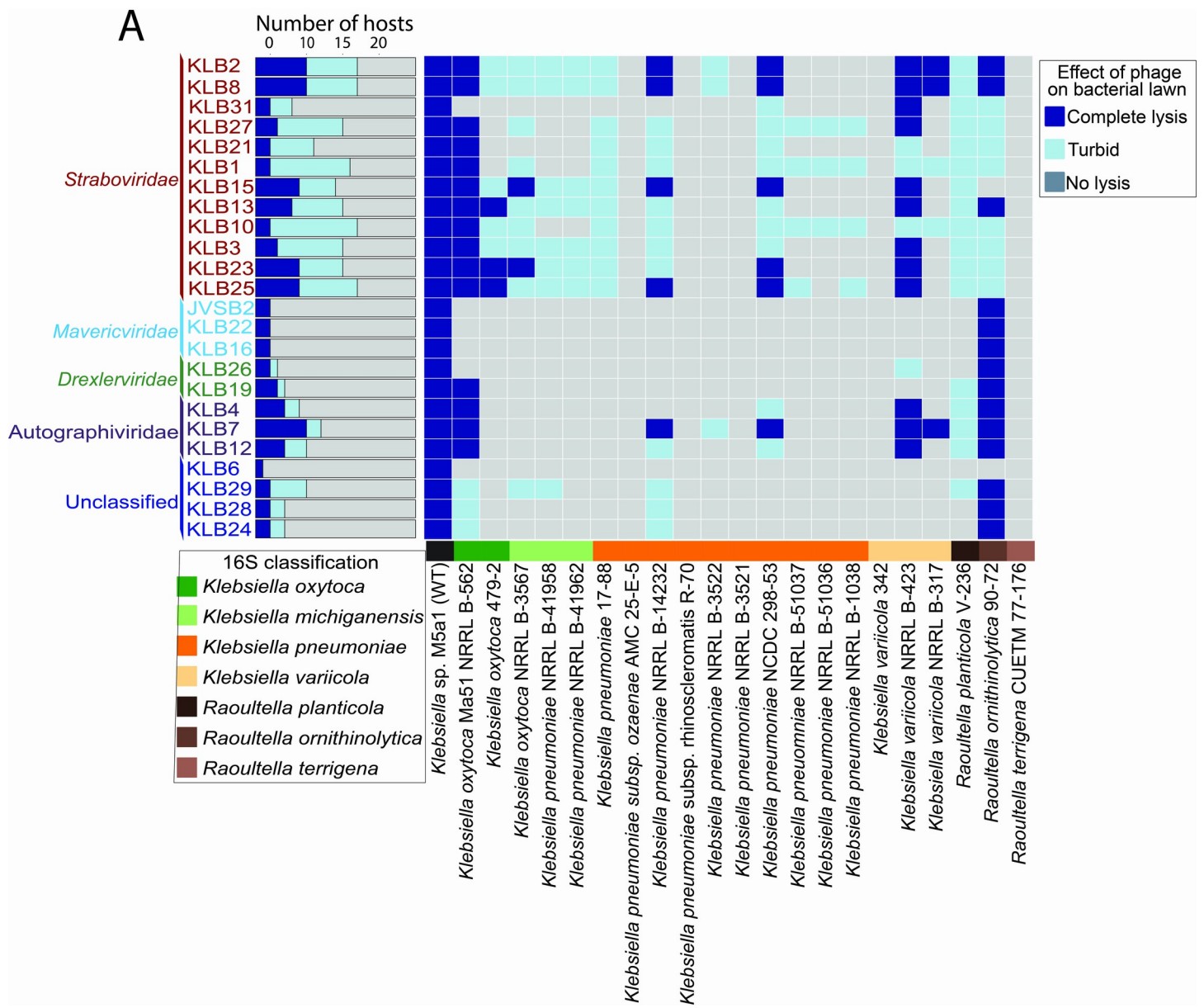

**Fig 7. Phage host-range testing of the 24 *Klebsiella* phages.** (A) Heatmap of the 24 *Klebsiella* phages screened against 19 *Klebsiella* isolates and 3 *Raoultella* isolates.

The phages in the *Straboviridae* phage family infected 5–14 isolates (average = 11.75). It has been previously found that phages in this family, specifically the *Slopekvirus* and the *Jiaoda-virus* genera, have broader host ranges, with many of these phages in this genus infecting >50% of *Klebsiella* isolates tested, including infecting several *Klebsiella* genera [7, 84, 86]. It is hypothesized that the broader host range of several phage genera in these groups could be due to homing endonucleases [86, 87] or tRNAs [43]. We compared the abundance of two types of homing endonucleases to the host range for our phages and the abundance of tRNAs to the host range. Still, we found no correlation between homing endonucleases ($R^2$ = 0.2979) or tRNA abundances ($R^2$ = 0.3908) and host range S8 Fig (Fig 7). Indicating in our *Slopekvirus*

phages tRNAs nor homing endonucleases significantly impacted the host range, and other factors are likely involved in driving host range.

KLB4, KLB7, and KLB12 in the *Autographivirdae* family had the second broadest host range and could infect between 6–9 strains, including 2 in the *Raoultella* genus. These *Autographiviridae* phages encoded 1–2 phage defense elements. KLB4, KLB7, and KLB12 encoded a S-adenosyl-L-methionine (SAM) hydrolase, and KLB4 encoded a dGTPase inhibitor. These phage defense elements protect the phage from type I restriction-modification systems [88] and depletion of the nucleotide pool [89], respectively, and may help explain the broader host range.

## Conclusion

Here, we isolated a diverse range of phages infecting soil-associated, plant growth-promoting, nonencapsulated *Klebsiella* sp. M5al. We found that many of these phages fell within the three largest families of *Klebsiella* phages. The isolation of these viruses led to the proposal of two novel phage families, Ca. *Mavericiviridae* and Ca. *Rivulusviridae*, which exemplifies the ever-expanding diversity and classification of known phages [41]. Identifying AMGs in the *Straboviridae* provides function-based hypotheses about phage and environment-specific metabolic reprogramming "niches" that parallel observations coming from focused "virocell" experimental systems [4, 90]. Host range analyses provide examples of phage families that are more specific versus general in their infection modalities, and as data scale across the field, will undoubtedly contribute to correlative inferences that will be predictive, a key advance for scaling phage therapy and other phage-based applications. Overall, this research significantly advances our understanding of *Klebsiella* phages infecting a soil-derived, nonencapsulated *Klebsiella* strain, which provides a foundation for in-depth "virocell" experiments to explore intricacies of phage biology, phage-bacteria interaction dynamics *in situ*, and ultimately applications in agriculture, environmental management, and biotechnology.

## Methods

### Sample collection for phage isolation

For phage isolation, soil, water, and sewage were collected around Ohio S1 Table. The soil was collected from personal properties by the owners. Water samples were collected from public rivers. For sewage, the influent was collected from the Jackson Pike and Southerly wastewater plants in Columbus, Ohio, with permission from the sewage plants. Influent samples collected from the Jackson Pike and Southerly WWTP's were provided by Dr. Thomas Wittum's laboratory. All samples were collected in Ohio, and phages were isolated at Ohio State University and Ashland University. No permits were required as all the sampling and phage collection were performed within Ohio state boundaries.

### Bacteriophage isolation

*Klebsiella* sp. M5al was kindly provided by Dr. Adam Arkin from the University of Berkeley and stored at -80˚C in 15% glycerol until use. *Klebsiella* sp. M5al was cultivated at 30˚C, shaking at 150 rpm in Luria Broth (LB). The *Klebsiella* phages were isolated from soil, water, and sewage samples around Ohio using a modified version of previously described techniques S1 Table [32]. Briefly, for soil samples, 1–2 grams of soil were resuspended into 8 mL of LB. For water samples, equal volumes of the water sample were mixed with 2XLB. For sewage samples, bacteria were first removed by spinning the samples at 8000 xg for 10 minutes to pellet debris. This was followed by filtration of the sewage through a 0.22 μM PES filter. The filtered sewage was mixed with equal volumes of 2XLB for a total volume of 9mLs. All samples were

supplemented with 10 mM CaCl$_2$ and 10 mM MgSo$_4$ and spiked with 100uL of an overnight culture of *Klebsiella* sp. M5al. These samples were then incubated overnight at 30˚C shaking at 150 rpm to enrich for phages infecting *Klebsiella* sp. M5al, then centrifuged at 8000xg for 10 minutes to pellet the bacteria. The lysate was filtered through a 0.22 μM PES filter to remove bacterial contaminants. Then 5 μL of the filtered lysates were spot plated onto a bacterial lawn consisting of 300 μL of overnight culture and 4 mL of 0.7% LB soft agar overlaid on a 1.4% LB agar plate. After 24 hours of incubation at 30˚C, spot plates were screened for zones of lysis. Lysates with positive lysis zones were serially diluted and then plated onto 4 mL of 0.7% LB soft agar overlay on a 1.4% LB agar plate with 300 μL of overnight bacterial culture added. Plates were incubated at 30˚C overnight. Single, isolated plaques were resuspended in 500 μL of SM buffer (100mM NaCl, 8mM MgSO4, 50mM Tris, pH7,5), diluted, and sequentially plated three times to obtain a monoclonal phage population. Phages were then amplified using previously described techniques [91].

## Phage DNA isolation and sequencing

Phage DNA was extracted as previously described [92]. Briefly, extracellular DNA and RNA were removed from $\sim 1^{10}$ PFU lysates and treated with DNase and RNase for 90 minutes. These enzymes were then deactivated using EDTA (final concentration 20 mM). Next, the proteinaceous phage capsids were degraded using Proteinase K (20mg/mL) and incubated at 56˚C for 90 minutes. The phage DNA was purified and concentrated using the DNeasy Blood and Tissue kit protocol for gram-negative bacteria for DNA purification (catalog #69506). Purified DNA was sequenced using NextSeq 2000 200Mbp at SeqCenter (previously MIGS), producing an average of 1.5 Mbp total read pairs per isolate.

## Phage genome assembly

Phage genomes were assembled as previously described [93, 94]. The reads were trimmed using trimmomatic (v.0.39.2) and checked for quality using fastQC (v.0.11.8) [95, 96]. Reads were subsampled using VSEARCH (v.2.14.1) to obtain a coverage between 20-200X. The phage genome was assembled by SPAdes (v.3.15.5) using subsampled reads and visualized using Bandage to ensure one complete contig was assembled [97–99]. The BBTools (v.38.69) package, BBmap, was used to check coverage at this step to ensure coverage was between 20-200X, as outlined in Millard's protocol [94, 100]. The output sam file from bbmap was converted to a bam file and indexed using samtools (v.1.10) (92). Error correction was performed by mapping the reads back to the assembled phage genome using Pilon (v.1.24) [101].

Phage termini were determined using several methods. First, all phage genomes and reads were run through PhageTerm (v.3.1.0) software to see if termini could be predicted using read mapping [102]. Phages that could not have their termini predicted were first checked for a circular genome permutation using the apc. pl script. This program also checks for repetitive artifacts from the assembly that were manually removed (https://github.com/jfass/apc/blob/master/apc.pl). Phages whose termini were not predicted via PhageTerm were then either (1) aligned based on the closest phage relative (>90% coverage*percent identity) or (2) for phages that did not have a close relative, these genomes were rearranged with an arbitrary gene like the small and large terminases S2 Table [93]. After the termini were determined for a phage, the coverage of all of the trimmed reads against the nearly finalized genome was checked with BBmap to determine the actual coverage [100], and the isolated genome underwent a final polishing step with Pilon to create the final genome assembly file [101].

## Long read sequencing of incomplete phage genomes

Three phages (KLB24, KLB28, KLB30) had incomplete contigs after short-read assembly with SPades. These phage genomes were sequenced using Oxford Nanopore Technology's MinION Mk1b long-read sequencer, and a hybrid, short, and long-read assembly was done to obtain a single, complete contig for each isolate. Briefly, reads were demultiplexed using the guppy basecaller (v2.3.1), and nanopore adapters were removed using porechop (v0.2.4) [103]. The BBtools reformat script removed reads shorter than 1000 base pairs [100]. Reads were then assembled using the hybrid option of SPAdes, which utilizes a short read assembly with correction by the long read sequences [98]. The subsampled short reads were used for the hybrid assembly. The same steps used in short read assembly are followed after hybrid SPAdes assembly of the reads.

## Phage genome annotation

Multiple tools were used to identify the ORFs in phage genomes with a scoring system that has been shown to provide the most accurate genome annotations [104, 105]. Our scoring system included eight ORF prediction tools GeneMark, GeneMarkS, GeneMarkS2, GeneMarkHMM, Glimmer, Prodigal, MetaGeneAnnotator, and Phanotate [106–113]. For each gene the scoring system used the number of ORF program calls, gene length, gene overlap, protein identification, programming potential, and presence in an operon. Scores of 3 or more were kept while scores of 0–2 were only kept if there was a BLAST hit or predicted by two or more ORF programs.

ORF functions were assigned based on high-scoring hits to BLASTp, Swiss-prot protein database, and HHPRED [51, 114]. Introns, spanins, terminators, and tRNAs were identified using programs from the structural and functional phage annotation pipelines in Apollo [115].

To calculate whether tRNA presence were due to codon biases, the codon usage for the bacteria and each phage was calculated in R using uco from the seqinr package. Codon usage of the bacterium was subtracted from the phage to calculate differences in codon usage. Boxplots were generated for each codon using ggplot2 [116].

## Phage taxonomy classifications

To identify the genus and family level of each phage, the complete genomes of all 24 phages were run through BLAST [51] to identify the closest phage relative. Using the top BLAST hit, the intergenomic similarity was calculated by multiplying the genome coverage by the percent identity. Using current ICTV recommendations, phages with ≥95% intergenomic similarity belonged to the same species and genus, while phages with ≥70% and <95% intergenomic similarity belonged to the same genus with a novel species [53].

The protein sequences of the 24 phages isolated in this study, the close relatives of those phages identified via BLAST, and all NCBI Virus RefSeq genomes (release 220) were run through a gene sharing network program (beta mode) [33] using default parameters S5 Table. With this gene-sharing network, distances are based on a Jaccard similarity-like distance metric, and instead of a single network, multiple networks at several clustering identities are employed. Optimal distance thresholds are identified for each network and each taxonomic rank from order to genus. By employing these disparate networks and adding group-specific markers, the gene-sharing network tool can classify realm, order, family, subfamily, and genus ranks with varying confidence levels for each realm. To find close proteomic relatives to the unclassified phages at the family level, we identified the RefSeq phages with the same phage family. KLB26 was classified in the *Drexlerviridae* phage family with a novel phage genus S5 Table. KLB24, KLB28, KLB29, and KLB5 were classified into a single phage family with 5

additional phages. JVSB2, KLB16, and KLB22 clustered in a single family with 334 additional phages. Non-RefSeq phages for the two novel phage families were identified via BLAST [51]. To define a phage family, we used ICTV recommended guidelines that these phages must (a) share a significant number of orthologues, (b) represent a cohesive and monophyletic group via proteome-based clustering tools, (c) if the phage family shares orthologues with another family, further confirm that these orthologues are a monophyletic lineage [53]. To confirm the phages formed a monophyletic lineage the phages from the two novel families were run through VipTree [52] and MMSeqs2 [66] to confirm that a significant number of genes were shared within the family.

Rivulusviridae formed a monophyletic lineage, while the Mavericviridae phages were poly-phyletic gene found in >95% of all phages S19 Table and S4 Fig. To be on the conservative side, we only used the phages identified in VipTree as monophyletic and had ∼10% of ORFs shared in >95% of all phages, so 239 phages in the Mavericviridae phage family were removed. All further analyses were on our 3 phages with the 92 phages that supported a monophyletic lineage.

## Phylogenetic analyses of phages

All Klebsiella infecting phages were retrieved from NCBI Virus using filters for viruses (taxid: 10239), nucleotide completeness, and Klebsiella (taxid: 570). As of February 6th, 2023, 581 phages in NCBI Virus were downloaded and used for this study [28]. We excluded 16 phages with 100% ANI, resulting in a final set of 565 Klebsiella phages.

To further reduce redundancy in the phage tree, these phages were clustered with MMseqs2 (60) using the options --min-seq-id 0.95 -c 0.8. This reduced the dataset to 523 sequences, which were subsequently used for the phylogenetic tree (Fig 1). For phylogenetic inference, we used two approaches. Firstly, we used a multi-loci gene tree comprising four genes: the large terminase, small terminase, major capsid, and portal gene. Multiple methods were used to extract the marker genes from every genome. Genes that were already annotated as the marker genes were extracted from the genomes in nucleotide format, and then each marker gene's list of proteins was manually inspected to remove obvious incorrect annotations, such as having too short or too long of a sequence compared to others. Each marker gene was clustered with VSEARCH using the --cluster_fast --id 0.75 options, and then every cluster was aligned using MAFFT, which was used to build a hidden Markov model (HMM) using hmmbuild with default options [117]. The HMM for each cluster of each marker gene was used to search a file containing every gene of the 537 phage dataset. The results were collated for each HMM of each marker gene, duplicates were removed, and results with an e-value lower than $10^{-3}$ were kept. If there were multiple results for a marker gene for one genome, the result with the lower e-value was kept. For the tree of all Klebsiella phages, the evolutionary model used was Blosum62+R5 for the large terminase, PMB+F+R3 for the small terminase, VT+F+I+G4 for the major capsid protein and the portal protein using the partition model and Maximum Likelihood phylogenetic inference in IQ-TREE 2.0 [118–120]. The phylogenetic trees were then visualized, and tree tips annotated and color-coded in iTOL v6 [121].

Phylogenetic inference of the Ca. Mavericviridae and the novel Ca. Rivulusviridae family were both based on a gene tree of the large terminase gene. Briefly, among representative reference NCBI sequences, the large terminase was extracted from the respective genomes based on their annotation and combined with large terminase genes annotated from our sequences. Sequences were then aligned using the E-INS-i strategy with over 1,000 iterations in MAFFT v7.017 [117]. All trees' aligned sequences were trimmed using Trimal [122] using the -gappy-out option that utilized gap and similarities distribution to establish the appropriate trimming

threshold. The trimmed sequence alignments were visualized and manually inspected in Jal-View [123], for overhangs and/or short sequences, which were subsequently removed. The most suitable evolutionary model for the large terminase sequences was inferred using Model-Finder [124]. The evolutionary model LG+I+G4 and LG+F+R5 was selected as the optimal evolutionary mode for the Ca. *Mavericviridae* and Ca. *Rivulusviridae*, respectively, and integrated into the Maximum Likelihood phylogenetic inference in IQ-TREE 2.0 [118].

## Comparison of phages against metagenomic datasets

Viral contigs were predicted from the Genome Resolved Open Watershed database (GROWdb) assemblies by first filtering contigs to only those larger than 5kb using seqkit version 2.7.0. Viral contigs >5kb were then predicted with Genomad v1.7.4 using the option "end-to-end" [125]. Viral contigs from all samples were collected into a single reference file for virus operational taxonomic units (vOTU) clustering and clustered using CheckV version 0.8.1 and a custom script that leverages BLAST+ with the scripts CheckV anicalc.py and aniclust.py with options "—min-ani 95 –min-tcov80" [126].

The prevalence of the 24 *Klebsiella* phages was mapped against vOTUs derived from the Global Soil Virus Atlas [34] and the GROWdb [35]. The 24 phage genomes were mapped against these vOTUs using CoverM v0.6. 1–3 using two options ----min-read-percent-identity .95 and --min-read-percent-identity .70 and --min-read-aligned-percent .75 --min-covered-fraction .70 -m trimmed_mean [127]. The depth of sequencing from each isolate then normalized coverage values. The reads from the 24 phage isolates were mapped against these vOTUs from both of the above datasets using CheckV's anicalc.py and aniclust.py with options "--min-ani 95 or --min-ani 70 and --min-tcov 80" [126].

## Phage genome comparisons

Phage genome comparisons were done using Clinker genome analyses [128]. The phage nucleotide sequences were uploaded to Clinker with minimum alignment sequence identity set to 0.3 [128]. One representative for each phage genus was used to compare nucleotide sequence to all other phages in the genus. The coloring of each gene was done manually based on product function. Stars denoting core genes identified using MMSeqs2.0 [66] were manually added to the Clinker maps using Adobe Illustrator.

## Identification of core genes in phages

The set of core genes for a single family was found using MMSeqs2. A database was made: mmseqs createdb prot.faa DB [66]. The database was clustered with a minimum sequence identity of 30% and a coverage of 80%: mmseqs cluster DB DB_clu --min-seq-id 0.30 -c 0.8 tmp. To analyze the data, a tsv file was made of the clustered database: mmseqs createtsv DB DB DB_clu mmseq_unchar_seq_0.3_cov_0.8.tsv. A gene was considered a core gene if present in >95% of the phages.

## Identification of AMGs

AMGs were identified using DRAM-v, the viral mode of the software DRAM [72]. The input was the finalized phage genome or a file with all the nonredundant *Klebsiella* phage sequences, and the output was the annotations made by DRAM-v S16 and 17 Tables. The annotations were curated so that only genes with a rank of C or higher were considered. Annotations with the flag M were considered AMGs, while AMGs containing V, B, or T flags were removed. Since our study looked at complete genomes and not metagenomic sequences that must worry

about prophage ends, the F flags were not removed, indicating that the gene was at the end of the genome. An overview of the functions of the AMGs in the manuscript was created using BioRender (Fig 6C). S7 Fig was generated in GraphPad. Statistical significance for S7A Fig statistical significance was calculated using a parametric, unpaired two-tail t-test. S7B and S7C Fig used linear regression to identify the $R^2$ value and tested the null hypothesis that the overall slope of the linear regression is 0, calculated from an F test.

## Host-range testing of phages

For host-range testing, each *Klebsiella* and *Raoultella* strain was incubated and grown at 30˚C 150 rpm overnight S18 Table. The bacterial lawn was made by mixing 300uL of the overnight *Klebsiella* or *Raoultella* strain with 5mL of 0.7% LB agar. Each phage was diluted to 1E8 PFU/ mL, and 5uL of the phage was plated onto a bacterial lawn. Plates were incubated at 30˚C overnight, and the phage spots were scored for either having a clear zone (complete lysis), a hazy zone of clearance (turbid), or no visual difference in the bacterial lawn (no lysis).

## Supporting information

**S1 Fig. tRNA codon bias in the *Straboviridae* phage family.** (A) cosine similarity of codons of each phage compared to *Klebsiella* sp. M5a1 for all genes, (B) genes in the late stage of phage infection, and (C) structural genes in the late stage of phage infection.
(TIF)

**S2 Fig. Representative *Straboviridae* clinker genome comparison.** Arrows represent forward (right) or reverse (left) open reading frames (ORFs), ORFs are color-coded by function, and ORFs shared across genomes are connected by shading that denotes their percent identity. Core ORFs shared across all RefSeq genomes in this family are marked with a black star. The genome figure was made using Clinker. The genome was split into four sections to better zoom in on the individual genes.
(TIF)

**S3 Fig. tRNA codon bias in the *Drexlerviridae* phage family.** (A) Cosine similarity of codons of each phage compared to *Klebsiella* sp. M5a1 for all genes, (B) genes in the late stages of phage infection, and (C) structural genes in the late stage of phage infection.
(TIF)

**S4 Fig. Viptree of the Ca. *Rivulusviridae* and Ca. *Mavericviridae* phage families compared to all Ref-Seq phages.** The red stars are the Ca. *Mavericidae* and Ca. *Rivulusviridae* phages.
(TIF)

**S5 Fig. VIRDIC genome analysis of the phages in the *Mavericviridae* family.** Phages with an intergenomic similarity ≥70% were grouped into a phage genus. Eleven phage genera of *Mavericviridae* are colored (green) *Bowlingvirus*, (dark blue) *Gwanakvirus*, (red) *Yanchengvirus*, (orange) *Alumvirus*, (purple) *Buckeyevirus*, (light green) *Ashvirus*, (light blue) *Kijivirus*, (yellow) *Tamuvirus*, (brown) *Auburnvirus*, (pink) *Hildvirus*, (light purple) *Dhillonvirus*. Phages labeled in white are the phages isolated in this study.
(TIF)

**S6 Fig. VIRDIC genome analysis of the phages in the *Rivulusviridae* phage family.** Phages with an intergenomic similarity ≥70% were grouped into a phage genus. 5 phage genera of *Rivulusviridae* are color-coded (green) *Colbvirus*, (blue) *Sherbvirus*, (yellow) *Lucvirus*, (orange) *Darbyvirus*, and (pink) *Cinnavirus*. Phages labeled in white are the phages isolated in this study.
(TIF)

**S7 Fig. Correlation of AMGs and genome size.** (A) Comparison of the genome sizes of phages that contain AMGs against phages that did not encode AMGs. A parametric, two-tailed, unpaired T-test was used to test statistical significance between the two groups. (B) Linear regression of phage containing AMGS compares genome size and the number of AMGs. (C) Linear regression of all *Klebsiella* phages genome sizes and number of AMGs. $R^2$ and statistical significance were calculated using standard settings of linear regression in GraphPad. All figures were made in GraphPad.
(TIF)

**S8 Fig. Comparison of the number of phage-encoded homing endonucleases and tRNA abundances to bacterial host range.** (A) Number of phage encoded homing endonucleases and number of bacteria a phage complete lyses and (B) completely or partially lyses. The number of phage-encoded tRNAs and the (C) number of bacteria a phage completely lyses and (D) completely or partially lyses. Linear regression of all *Klebsiella* phage genome sizes and number of AMGs. $R^2$ and statistical significance were calculated using standard settings of linear regression in GraphPad. All figures were made in GraphPad.
(TIF)

**S9 Fig. VIRDIC genome analysis of all the phages isolated in this study.**
(TIF)

**S1 Table. Information on sampling sites for phages isolated in this study.** Information on the sample site, type of sample, date of sampling, date of enrichment, and city that each phage was isolated from.
(XLSX)

**S2 Table. General information for phages isolated in this study.**
(XLSX)

**S3 Table. Top BLAST results for each phage.** The top BLAST result for each phage includes the query coverage, e value, percent identity, genome accession, and taxonomy. The intergenomic similarity was calculated by multiplying the query coverage by the percent identity and taxonomy for isolated phages using current ICTV guidelines.
(XLSX)

**S4 Table. MMSeqs2 clustering of *Drexlerviridae* RefSeq phages.**
(XLSX)

**S5 Table. vConTACT3.0 clustering of Ca. *Mavericviridae* and Ca. *Rivulusviridae* phages.**
(XLSX)

**S6 Table. Similarity hits of the *Klebsiella* phages to the GSVA and GROWdb viruses.**
(XLSX)

**S7 Table. Accession of all non-redundant *Klebsiella* phages.**
(XLSX)

**S8 Table. List of RefSeq phages in the *Straboviridae* family.**
(XLSX)

**S9 Table. List of RefSeq phages in the *Drexlerviridae* family.**
(XLSX)

**S10 Table. List of RefSeq phages in the *Autographiviridae* family.**
(XLSX)

**S11 Table. MMSeqs2 clustering of the 95 Ca. *Mavericviridae* phages.**
(XLSX)

**S12 Table. MMSeqs2 clustering of the Ca. *Mavericviridae* and *Drexlerviridae* RefSeq phages.**
(XLSX)

**S13 Table. MMSeqs2 clustering of the Ca. *Rivulusviridae* RefSeq phages.**
(XLSX)

**S14 Table. MMSeqs2 clustering of the *Straboviridae* RefSeq phages.**
(XLSX)

**S15 Table. MMSeqs2 clustering of the *Autographiviridae* RefSeq phages.**
(XLSX)

**S16 Table. Dram-v output of the auxiliary metabolic genes from the NCBI *Klebsiella* phages.**
(XLSX)

**S17 Table. Dram-v output of the auxiliary metabolic genes from the 24 phages isolated in this study.**
(XLSX)

**S18 Table. List of bacterial strains used in this study.**
(XLSX)

**S19 Table. MMSeqs2 clustering of the 334 phages identified with vConTACT3.0 and the three *Mavericviridae* phages isolated in this study.**
(XLSX)

## Acknowledgments

We want to thank the Center of Microbiome Science at OSU for their bioinformatic support in calculating the occurrence of our phages in the GSVA and GROWdb datasets. We also thank Marie Burris for the soil and water samples used for phage isolation and Funing Tian for her feedback on the manuscript. We thank Brian Ahmer for the *Klebsiella* and *Raoultella* isolates, as well as the USDA-ARS Culture Collection (NRRL) for several microbial strains used in this work.

## Author Contributions

**Conceptualization:** Marissa R. Gittrich, Vivek K. Mutalik, Paul Hyman.

**Data curation:** Marissa R. Gittrich, Courtney M. Sanderson, Cara M. Noel, Jonathan E. Leopold, Sumeyra C. Selbes, Olivia R. Farinas, Jack Caine, Joshua Davis II.

**Formal analysis:** Marissa R. Gittrich, Courtney M. Sanderson, James M. Wainaina, Cara M. Noel, Jonathan E. Leopold, Erica Babusci.

**Funding acquisition:** Vivek K. Mutalik, Paul Hyman, Matthew B. Sullivan.

**Investigation:** Marissa R. Gittrich, Cara M. Noel, Jonathan E. Leopold, Erica Babusci, Sumeyra C. Selbes, Olivia R. Farinas, Jack Caine, Joshua Davis II.

**Methodology:** Marissa R. Gittrich, Courtney M. Sanderson, James M. Wainaina, Paul Hyman.

**Project administration:** Marissa R. Gittrich, Vivek K. Mutalik.

**Resources:** Marissa R. Gittrich, Paul Hyman.

**Supervision:** Vivek K. Mutalik, Matthew B. Sullivan.

**Validation:** Marissa R. Gittrich, Vivek K. Mutalik.

**Visualization:** Marissa R. Gittrich, Courtney M. Sanderson, James M. Wainaina.

**Writing – original draft:** Marissa R. Gittrich, Vivek K. Mutalik, Paul Hyman, Matthew B. Sullivan.

**Writing – review & editing:** Marissa R. Gittrich, Courtney M. Sanderson, James M. Wainaina, Cara M. Noel, Jonathan E. Leopold, Erica Babusci, Sumeyra C. Selbes, Olivia R. Farinas, Vivek K. Mutalik, Paul Hyman, Matthew B. Sullivan.

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
