## [Decision Letter · Decision Letter 0]

6 Aug 2024

PONE-D-24-24676Isolation and characterization of 24 phages infecting the plant growth-promoting rhizobacterium Klebsiella sp. M5alPLOS ONE

Dear Dr. Sullivan,

Thank you for submitting your manuscript to PLOS ONE. After careful consideration, we feel that it has merit but does not fully meet PLOS ONE’s publication criteria as it currently stands. Therefore, we invite you to submit a revised version of the manuscript that addresses the points raised during the review process.

We look forward to receiving your revised manuscript.

Kind regards,

Erika Kothe

Academic Editor

PLOS ONE

4. Please amend the manuscript submission data (via Edit Submission) to include authors Courtney M. Sanderson, James M. Wainaina, Cara M. Noel, Jonathan E. Leopold, Erica Babusci, Sumeyra C. Selbes, Olivia R. Farinas, Jack Caine, Joshua Davis II, Vivek K. Mutalik and Paul Hyman. 

Additional Editor Comments:

Both reviewers recommend changes to the manuscript. I agree that additional information on the phages including, e.g., electron micrographs, would be required. Other than that, please also carefully consider the further comments by both reviewers.

Reviewers' comments:

Reviewer's Responses to Questions

**Comments to the Author**

1. Is the manuscript technically sound, and do the data support the conclusions?

Reviewer #1: Yes

Reviewer #2: Yes

2. Has the statistical analysis been performed appropriately and rigorously? 

Reviewer #1: Yes

Reviewer #2: N/A

3. Have the authors made all data underlying the findings in their manuscript fully available?

Reviewer #1: Yes

Reviewer #2: Yes

4. Is the manuscript presented in an intelligible fashion and written in standard English?

Reviewer #1: Yes

Reviewer #2: Yes

5. Review Comments to the Author

Reviewer #1: Soil is an important source of food and medicine, thus is vital for both people and the planet. However, model systems for soil virus studies are currently lacking. Klebsiella sp. M5al is a suitable candidate which can be used as a model strain to study the virus-host interactions in the rhizosphere. In this manuscript, Gittrich et al. reported the isolation of 24 phages that infect Klebsiella sp. M5al from soil, water, and sewage environmental samples. They sequenced the whole genomes of these phages, conducted systematic comparative genomic, taxonomic, and phylogenetic analyses, and identified auxiliary metabolic genes. They also examined the host ranges of these phages by using 22 bacterial strain panel that included 4 Klebsiella species and 3 Raoultella species. Overall, this work provides important pure culture systems for studying the viral functions and phage-host interactions in soil environments, especially the rhizosphere. In general, the manuscript was well-organized and readable. I only have a major concern and some minor comments are listed as follows.

Major:

Morphology is the basic information of isolated phages. In my opinion, the authors should at least provide some transmission electron microscope graphs of representative phages among the 24 isolated phages. Additionally, since many soil metagenomes and viromes have been published, I’m curious about the prevalence and abundance of these isolated phages in the soil environments. The authors do not need to perform additionally analysis, but may consider adding some related discussion or outlook in the manuscript, just a suggestion if it is reasonably possible.

Minor:

1, The units (bp) of genome size in Table 1 should be specified.

2, There is a typo in Fig. 2A, “Jiaoadavirus” should be corrected to “Jiaodavirus”.

3, Fig. 2B, The genome comparisons of Straboviridae is difficult to see clearly for details, especially for those genes with small size.

4, Fig. 2B, “Identity” should be specifically indicated whether it is based on DNA or amino acid sequence comparison.

5, Fig. 4B, “UFBootstrap support is shown as a circle on each node”. Like other figures, the Bootstrap value range represented by the circle should be indicated.

6, Fig. S6, units should be in brackets, for example, genome size (bp).

Reviewer #2: The authors have completed an extensive task sequencing and characterizing 24 novel phage genomes. The manuscript is well written and includes a comprehensive methods section. I only have two minor comments.

1. In table 1, I recommend the authors to put all the phages of the same family together. It makes comparing within the family much easier.

2. I was not able to access any phage genomes with the provided accession numbers. Please make sure these genomes are publicly available before the manuscript is published.

3. I would also like to some more discussion about the potential ecological role of these phage other than the AMG part.

Overall, a good manuscript.

6. PLOS authors have the option to publish the peer review history of their article (what does this mean?). If published, this will include your full peer review and any attached files.

Reviewer #1: No

Reviewer #2: **Yes: **Chinmay V Tikhe

---

## [Author Response · Author response to Decision Letter 0]

15 Oct 2024

Dear Erika Kothe,

Thank you for considering our manuscript for publication in PLOS ONE and for providing valuable feedback to help improve and strengthen our work. We greatly appreciate the thoughtful comments and suggestions from you and the two reviewers. Please find our point-by-point responses below, highlighted in blue for clarity, and attached the revised manuscript. We believe these revisions have significantly improved the manuscript and hope it now meets the standards for publication in PLOS ONE.

Response: This manuscript was formatted for PLOS Biology and transferred from that journal. We have now updated the titles for each section, resized the figure files, and updated all file names to fit the requested requirements for PLOS One.

Response: Thank you for flagging this additional information should be included. We have now added a section in the methods describing the sample collection and permissions for the soil, water, and sewage samples. Notably, we understand that there are no permits required as everything was collected from one of three scenarios: (a) from the owners of the property and those owners are authors or are acknowledged in the paper, (b) from public rivers where no permit was required, or (c) from a sewage plant with permission from the plant. These samples were collected and processed within Ohio and thus did not require transport permits.

Response: Funding information, financial disclosure awards, and grant numbers have been changed accordingly.

4. Please amend the manuscript submission data (via Edit Submission) to include authors Courtney M. Sanderson, James M. Wainaina, Cara M. Noel, Jonathan E. Leopold, Erica Babusci, Sumeyra C. Selbes, Olivia R. Farinas, Jack Caine, Joshua Davis II, Vivek K. Mutalik and Paul Hyman. 

Response: All authors have now been included in the manuscript submission data.

5. Review Comments to the Author

REVIEWER #1:

Soil is an important source of food and medicine, thus is vital for both people and the planet. However, model systems for soil virus studies are currently lacking. Klebsiella sp. M5al is a suitable candidate which can be used as a model strain to study the virus-host interactions in the rhizosphere. In this manuscript, Gittrich et al. reported the isolation of 24 phages that infect Klebsiella sp. M5al from soil, water, and sewage environmental samples. They sequenced the whole genomes of these phages, conducted systematic comparative genomic, taxonomic, and phylogenetic analyses, and identified auxiliary metabolic genes. They also examined the host ranges of these phages by using 22 bacterial strain panel that included 4 Klebsiella species and 3 Raoultella species. Overall, this work provides important pure culture systems for studying the viral functions and phage-host interactions in soil environments, especially the rhizosphere. In general, the manuscript was well-organized and readable. I only have a major concern and some minor comments are listed as follows.

Major:

Morphology is the basic information of isolated phages. In my opinion, the authors should at least provide some transmission electron microscope graphs of representative phages among the 24 isolated phages. 

Response: Transmission electron microscopy was the gold standard for decades due to TEM being required to define the phage family based on the phages’ morphological characteristics. However, with advances in DNA sequencing and increased complete phage genomes publicly available, the International Committee on the Taxonomy of Viruses (ICTV) has moved away from electron micrograph based morphology due to morphology-defined families being evolutionarily polyphyletic. Importantly, in 2022, the ICTV abolished polyphyletic morphological families Siphoviridae, Myoviridae, and Podoviridae and instead now uses genome-based standards for family identification using the sequenced genomes and comparing core genes, genetic content, and evolutionary trees to define phage families [1]. Since 2022, electron microscopy is often still included in phage characterization papers, but its importance is greatly diminished. For example, where such data are included the information might represent a single sentence of description, but no further information or systems-level understanding is gained beyond stating its morphology [2–4]. In other casts, such information is not included at all, and these are cases where such morphological information had little value for the questions driving those researchers’ science [5–7]. In our case, we have chosen not to establish micrographs because our future research does not require morphological characterization, and we are neither set up nor funded to obtain such electron micrographs for these isolates and, in many cases, at least high-level morphological predictions can be made from the genomes themselves. 

However, we appreciate that other research questions might require morphological understanding. To facilitate this, we have placed these phages into long-term storage in our lab (available upon request), as well as deposited them in an international phage repository (the Félix d'Hérelle Reference Center for bacterial viruses of the Université Laval) that will help make the phages broadly available to other researchers interested in their morphological characteristics to be able to work with the phages directly. 

Additionally, since many soil metagenomes and viromes have been published, I’m curious about the prevalence and abundance of these isolated phages in soil environments. The authors do not need to perform additional analysis but may consider adding some related discussion or outlook in the manuscript; this is just a suggestion if it is reasonably possible.

Response: Thank you for the suggestion. We were also interested, but at submission there were few reasonable databases to conduct such a search. However, serendipitously, two large-scale relevant datasets have emerged since then that take separately each a global soils and US rivers metagenomic view of phages. Using these two datasets, we assessed whether our phage genomes were identified at the species, genus, and family level. The results are underwhelming in that our phages were not found or extremely rare in these datasets; however, they support the hypothesis already in the literature that soil phages are drastically under-sampled to date and thus rarely observed with any broad biogeography at all. We now report on these results in the section on the prevalence of the isolated phages in soils and rivers and updated methods with a description of these analyses.

Minor:

1, The units (bp) of genome size in Table 1 should be specified.

Response: We added (bp) to the genome size in Table 1.

2, There is a typo in Fig. 2A, “Jiaoadavirus” should be corrected to “Jiaodavirus”.

Response: Typo in Fig. 2A was fixed.

3, Fig. 2B, The genome comparisons of Straboviridae is difficult to see clearly for details, especially for those genes with small size.

Response: We apologize for the oversight. Straboviridae comparisons are difficult due to the relatively large genomes of 160-180kb genomes compared to the other dsDNA phages that are ~60-70kb. To address this challenge and for ease of the reader, we have 

added a supplemental figure with the genome divided into four sections this should enable 

an in-depth look at certain sections of the Straboviridae phage genome

.

4, Fig. 2B, “Identity” should be specifically indicated whether it is based on DNA or amino acid sequence comparison.

Response: We thank the reviewer for highlighting this, we have added amino acid identity to that figure to show that it was a protein comparison. In addition, we have also added that to all other figures that have CLINKER images.

5, Fig. 4B, “UFBootstrap support is shown as a circle on each node”. Like other figures, the Bootstrap value range represented by the circle should be indicated.

Response: The UFBootstrap support was added to Figure 4B

6, Fig. S6, units should be in brackets, for example, genome size (bp).

Response: Brackets were added to Fig S6 to the axis’s that have bp or kbp

REVIEWER #2:

The authors have completed an extensive task sequencing and characterizing 24 novel phage genomes. The manuscript is well written and includes a comprehensive methods section. I only have two minor comments.

1. In table 1, I recommend the authors to put all the phages of the same family together. It makes comparing within the family much easier.

Response: Thank you for the suggestion we reordered Table 1 by family, and by the order we describe each phage family in the text.

2. I was not able to access any phage genomes with the provided accession numbers. Please make sure these genomes are publicly available before the manuscript is published.

Response: The phage genomes were submitted to GenBank in April 2024, where they then get reviewed for discrepancies in their annotation, and the accession numbers are to be made available upon publication of the manuscript.

3. I would also like to some more discussion about the potential ecological role of these phage other than the AMG part.

Response: Thank you for this comment. Given a similar request also from reviewer #1, we have made a very large effort to assess ecological prevalence and abundance of the phages through comparison to two large-scale, recently available global soil virus and national river virus datasets. As described above, we used these datasets to assess whether our phage genomes were identified at the species, genus, and family level. The results are underwhelming in that our phages were not found or extremely rare in these datasets; however, they support the hypothesis already in the literature that soil phages are drastically under-sampled to date and thus rarely observed with any broad biogeography at all. We now report on these results in the section on the prevalence of the isolated phages in soils and rivers and updated methods with a description of these analyses. 

Overall, a good manuscript.

Response: Thank you for the kind words, and we appreciate your time reviewing.

References

1. Turner D, Shkoporov AN, Lood C, Millard AD, Dutilh BE, Alfenas-Zerbini P, et al. Abolishment of morphology-based taxa and change to binomial species names: 2022 taxonomy update of the ICTV bacterial viruses subcommittee. Arch Virol. 2023;168(2):74. 

2. Rackow B, Rolland C, Mohnen I, Wittmann J, Müsken M, Overmann J, et al. Isolation and characterization of the new Streptomyces phages Kamino, Geonosis, Abafar, and Scarif infecting a broad range of host species. Microbiol Spectr. 2024 Sep 25;0(0):e00663-24. 

3. Sada TS, Tessema TS. Isolation and characterization of lytic bacteriophages from various sources in Addis Ababa against antimicrobial-resistant diarrheagenic Escherichia coli strains and evaluation of their therapeutic potential. BMC Infect Dis. 2024 Mar 14;24(1):310. 

4. Feltin C, Garneau JR, Morris CE, Bérard A, Torres-Barceló C. Novel phages of Pseudomonas syringae unveil numerous potential auxiliary metabolic genes. J Gen Virol. 2024 Jun;105(6):001990. 

5. Lourenço M, Osbelt L, Passet V, Gravey F, Megrian D, Strowig T, et al. Phages against Noncapsulated Klebsiella pneumoniae: Broader Host range, Slower Resistance. Microbiol Spectr. 2023 Jun 20;0(0):e04812-22. 

6. Finney AG, Perry JM, Evans DR, Westbrook KJ, McElheny CL, Iovleva A, et al. Isolation and Characterization of Lytic Bacteriophages Targeting Diverse Enterobacter spp. Clinical Isolates. PHAGE. 2022 Mar;3(1):50–8. 

7. Fletcher J, Manley R, Fitch C, Bugert C, Moore K, Farbos A, et al. The Citizen Phage Library: Rapid Isolation of Phages for the Treatment of Antibiotic Resistant Infections in the UK. Microorganisms. 2024 Feb;12(2):253.

---

## [Decision Letter · Decision Letter 1]

4 Nov 2024

Isolation and characterization of 24 phages infecting the plant growth-promoting rhizobacterium Klebsiella sp. M5al

PONE-D-24-24676R1

Dear Dr. Sullivan,

We’re pleased to inform you that your manuscript has been judged scientifically suitable for publication and will be formally accepted for publication once it meets all outstanding technical requirements.

Kind regards,

Erika Kothe

Academic Editor

PLOS ONE

Additional Editor Comments (optional):

Reviewers' comments:

Reviewer's Responses to Questions

**Comments to the Author**

1. If the authors have adequately addressed your comments raised in a previous round of review and you feel that this manuscript is now acceptable for publication, you may indicate that here to bypass the “Comments to the Author” section, enter your conflict of interest statement in the “Confidential to Editor” section, and submit your "Accept" recommendation.

Reviewer #1: All comments have been addressed

2. Is the manuscript technically sound, and do the data support the conclusions?

Reviewer #1: Yes

3. Has the statistical analysis been performed appropriately and rigorously? 

Reviewer #1: Yes

4. Have the authors made all data underlying the findings in their manuscript fully available?

Reviewer #1: Yes

5. Is the manuscript presented in an intelligible fashion and written in standard English?

Reviewer #1: Yes

6. Review Comments to the Author

Reviewer #1: The authors have adequately addressed my concerns. I have no further comments, and congratulate the authors for a very interesting study.

7. PLOS authors have the option to publish the peer review history of their article (what does this mean?). If published, this will include your full peer review and any attached files.

Reviewer #1: No

---

## [Editor Report · Acceptance letter]

15 Nov 2024

PONE-D-24-24676R1 

PLOS ONE

Dear Dr. Sullivan, 

I'm pleased to inform you that your manuscript has been deemed suitable for publication in PLOS ONE. Congratulations! Your manuscript is now being handed over to our production team.

Kind regards, 

on behalf of

Prof. Dr. Erika Kothe 

Academic Editor

PLOS ONE